



**Chemical properties, sources and size-resolved**
**hygroscopicity of submicron black carbon-containing**
**aerosols in urban Shanghai**
Shijie Cui[1], Dan Dan Huang[2], Yangzhou Wu[1,a], Junfeng Wang[1], Fuzhen Shen[1,b], Jiukun
Xian[1], Yunjiang Zhang[1], Hongli Wang[2], Cheng Huang[2], Hong Liao[1], Xinlei Ge[1, *]
[1] Jiangsu Key Laboratory of Atmospheric Environment Monitoring and Pollution
Control, Collaborative Innovation Center of Atmospheric Environment and Equipment
Technology, School of Environmental Science and Engineering, Nanjing University of
Information Science and Technology, Nanjing 210044, China
[2] Shanghai Academy of Environmental Sciences, Shanghai 200233, China
[a] now at: Department of Atmospheric Sciences, School of Earth Sciences, Zhejiang
University, Hangzhou 310027, PR China
[b] now at: Department of Meteorology, University of Reading, Reading, RG6 6BX, UK
*Corresponding author: Xinlei Ge (Email: caxinra@163.com)
For *Atmospheric Chemistry and Physics*



**Abstract.** Refractory black carbon ($r$BC) aerosols play an important role in air quality
and climate change, yet high time-resolved and detailed investigation on the
physicochemical properties of $r$BC and its associated coating is still scarce. In this work,
we used a laser-only Aerodyne soot particle aerosol mass spectrometer (SP-AMS) to
exclusively measure the $r$BC-containing ($r$BCc) particles, and compared their
properties with the total non-refractory submicron particles (NR-PM$_1$) measured in
parallel by a high-resolution AMS (HR-AMS) in Shanghai. The observation shows that
$r$BC was overall thickly coated with an average mass ratio of coating to $r$BC core ($R_{BC}$)
of ~5.0. However, mass of $r$BC coating species only occupied 19.1% of those in NR-
PM$_1$; sulfate tended to condense preferentially on non-$r$BC particles therefore its
portion on $r$BC was only 7.4%, while the majority of primary organic aerosols (POA)
were associated with $r$BC (72.7%). Positive matrix factorization reveals that cooking
emitted organics did not coat on $r$BC, and a portion of organics coated on $r$BC was
from biomass burning which was unidentifiable in NR-PM$_1$ organics. Small $r$BCc
particles were predominantly from traffic, while large-sized ones were often mixed with
secondary components and typically had thick coating. During this campaign, sulfate
and secondary organic aerosol (SOA) species were generated mainly through daytime
photochemical oxidation (SOA formation likely involved with in-situ chemical
conversion of traffic-related POA to SOA), while nocturnal heterogeneous formation
was dominant for nitrate; we also estimated the average times of 5~19 hours for those
secondary species to coat on $r$BC. Particles during a short period that was affected by
ship emissions, were characterized with a high vanadium concentration (on average 5.8
ng m$^{-3}$) and a vanadium/nickel mass ratio of 2.0. Furthermore, the size-resolved
hygroscopicity parameter ($\kappa_{r\mathrm{BCc}}$) of $r$BCc particles was obtained based on its fully
chemical characterization, and was parameterized as $\kappa_{r\mathrm{BCc}}(x)= 0.29-0.14 \times \exp(-0.006$
$\times x)$ ($x$ is from 150 to 1000 nm). Under critical supersaturations ($SS_C$) of 0.1% and 0.2%,
the $D_{50}$ values were $166 \pm 16$ and $110 \pm 5$ nm, respectively, and with $16 \pm 3\%$ and $59 \pm$
4% of $r$BCc in number could be activated into cloud condensation nuclei (CCN). Our
findings are valuable to advance the understanding of BC chemistry as well as the
effective control of atmospheric BC pollution.



## 1 Introduction

Refractory black carbon ($r$BC) aerosols can directly absorb solar radiation, indirectly change the nature of cloud and alter the albedo of snow or glaciers (Jacobi et al., 2015), resulting in a positive radiative forcing that is second only to carbon dioxide on both regional and global scales (Ramanathan and Carmichael, 2008;Bond et al., 2013). The fresh $r$BC particles produced by incomplete combustion of biomass and fossil fuel tend to be fractal in morphology and can mix with many other components (Peng et al., 2016;Li et al., 2021). After entering into the atmosphere, fresh $r$BC can further externally or internally mix with organic/inorganic species which are primarily emitted or secondarily formed, and such aged $r$BC-containing ($r$BCc) particles (Chen et al., 2017;Lee et al., 2017) might have contrasting chemical properties and morphologies (or mixing states) (Liu et al., 2017a;Lee et al., 2019;Xie et al., 2019). In addition, when $r$BC mixes with hydrophilic materials, its hygroscopicity, cloud condensation nuclei (CCN) activity and size distribution, etc., can be significantly changed, which subsquently affect its atmospheric behavior, impact and lifecycle (Liu et al., 2013;Lambe et al., 2015). Therefore, it is necessary to elucidate the physicochemical characteristics and sources of $r$BC cores and associated coating materials, so as to better understand their influences on climate and air quality.

Chemical composition of ambient $r$BCc particles is largely dependent upon atmospheric conditions and emission sources. In general, the thickness of coating, mass contribution of secondary components (such as sulfate, nitrate and secondary organic aerosol (SOA) species) and oxidation degree of the coated organics of $r$BCc particles, increase with the aging time or oxidation capacity of ambient environment (Cappa et al., 2012;Liu et al., 2015;Wang et al., 2017;Collier et al., 2018;Wang et al., 2019), except in some specific cases that thickly coated $r$BCc might be dominated by primarily emitted particles (such as from biomass burning (Wang et al., 2017)). Recent field observations report that SOA species coated on $r$BC cores could account for 35% and 41% of the total SOA mass near traffic emission sources and in a polluted offshore environment, respectively (Massoli et al., 2012;Massoli et al., 2015). A study of $r$BCc particles in Singapore finds that over 90% of $r$BC derived from local combustion



sources (mainly traffic), while 30% of $r$BC was associated with fresh SOA generated
under the influences of daytime shipping and industrial emissions (Rivellini et al.,
2020).The SOA material concentrated on the surface of $r$BC was found to be chemically
different from the SOA that was externally mixed with $r$BC (Lee et al., 2017) in Fontana,
California, and another study in Shenzhen, China, reveals that more oxidized SOA
preferred to mix with $r$BC due to that  abundant transition metals detected on $r$BC cores
might act as catalysts to convert less oxidized SOA to more oxidized SOA in aerosol
aqueous phase (Cao et al., 2022); the $r$BC could catalyze $SO_2$ to form sulfate as well,
as observed in Beijing (Zhang et al., 2020) and Guangzhou (Zhang et al., 2021), China.
Besides SOA, cooking-related OA is found to be externally mixed with $r$BC (Lee et al.,
2017;Wang et al., 2019), and a unique biomass burning related OA factor  was identified
and was only present in $r$BCc rather than non-$r$BC particles during summertime in
Beijing (Wang et al., 2020a).
Size distribution of $r$BCc particles is also modulated greatly by their original
sources and ageing processes. For example, a study in Shanghai shows a bimodal size
distribution of $r$BCc, with a condensation mode dominated by traffic emissions (small
core size, thin coating) and a droplet mode including highly aged biomass burning
particles (large core size and thick coating) and highly aged traffic particles (small core
size and very thick coating) (Gong et al., 2016). Another study in Beijing (Liu et al.,
2019) further resolves four size modes of $r$BCc, relevant with traffic (small core, thin
coating), coal or biomass burning (moderate coating, both small and large cores), coal
combustion (large core, think coating) and secondary process (thick coating, both small
and large cores).
Moreover, water uptake and CCN activity of $r$BCc particles can increase with the
encapsulation of water-soluble substances such as sulfate, nitrate, and SOA (Liu et al.,
2013;Wu et al., 2019). Based on the measured chemical composition of $r$BCc, our
previous work has established a method for calculating size-resolved hygroscopicity
parameters of $r$BCc ($\kappa_{r\mathrm{BCc}}$), and determined the CCN activation diameters of $r$BCc
particles for given critical supersaturation ($SS_C$) values (Wu et al., 2019).
Highly time-resolved chemical characterization of $r$BCc particles were seldom



reported in China and is still lacking in Shanghai. In this study, we utilized an Aerodyne
soot particle aerosol mass spectrometer (SP-AMS) to determine the concentration,
composition and size distribution of $r$BCc particles exclusively (technical details in
Section 2.1) in urban Shanghai for the first time. We also compared the SP-AMS
measurement results with those from a co-located Aerodyne high-resolution time-of-
flight aerosol mass spectrometer (HR-AMS), to comprehensively investigate the
characteristics of $r$BCc particles. We analyzed $\kappa_{r\mathrm{BCc}}$ and estimated the proportions of
activated $r$BCc numbers at given $SS_\mathrm{C}$ as well.

**2 Experimental methods**
**2.1 Sampling site and instrumentation**

The field measurement was conducted from October 31 to December 2, 2018,

during which the instruments were deployed on 8th floor of the building of Shanghai
Academy of Environmental Sciences (SAES) (31°10'33.348" N, 121°26'10.978" E).
East of the sampling site is a large commercial shopping center, and the site is
surrounded by residential areas with two busy arterial roads directly to the east (~450
m) and south (~150 m), respectively (Figure S1 in the supporting information). In
addition, the adjacent areas are densely populated with roadside residents, office
workers, and market traders, as well as crowds in and out of the Caobao Road Metro
station (~100 m). The measurement period was dominated by northeasterly winds,
while many international freight companies located on northeastern side of the site, and
many freighters were reposing on the Huangpu River. Overall, the sampling site was
probably influenced by vehicular emissions, residential activities and the northeast
Cargo ship emission plumes, etc.

An Aerodyne SP-AMS and an HR-AMS were operated in parallel during the

campaign. The two AMSs shared a same sampling line with a PM$_{2.5}$ cyclone (Model
URG-2000-30EN) in front to remove coarse particles. Ambient air pulled through the
sampling line was dried using a diffusion dryer filled with silicon gel and was
subsequently drawn into both instruments. Due to the transmission efficiency of the
inlet lens, both AMSs measured mainly particles of 30-1200 nm (denoted as PM$_1$).


The working principle of SP-AMS has been described in detail previously (Onasch
et al., 2012). However, in this work, we used only the intracavity infrared laser
vaporizer to selectively measure $r$BCc particles ($r$BC cores and associated coating
materials), as $r$BC can absorb 1064 nm laser light. The thermal tungsten vaporizer had
to be physically detached otherwise non-$r$BC particles can still be detected as the
filament can heat the vaporizer to ~200 °C even if it was turned off. Before sampling,
the SP-AMS was tuned and calibrated following the steps described previously (Lee et
al., 2015;Willis et al., 2016;Wang et al., 2017). During sampling, due to relatively low
$r$BCc mass loadings, the SP-AMS was operated with two mass sensitive V modes (2.5
minutes per cycle), one with a particle time-of-flight (PToF) mode (30 s) and another
one (120 s) with a mass spectral mode with mass-to-charge ($m/z$) ratio up to 500.
Filtered air was also measured in the middle of campaign (for 60 min) to determine the
limits of detection (LOD, three times the standard deviation) of various aerosol species
and to adjust the air-influenced mass spectral signals (Zhang et al., 2005).
Before removal of the tungsten heater, the calibrations of ionization efficiency (IE)
for nitrate and relative ionization efficiency (RIE) of sulfate were performed by using
pure ammonium nitrate and ammonium sulfate particles (Jayne et al., 2000), and the
values were assumed to be unchanged throughout the whole campaign (Willis et al.,
2016). RIE of $r$BC to nitrate was calibrated by using size-selected (300 nm) BC
particles (REGAL 400R pigment black, Cabot Corp.) (Onasch et al., 2012), and the
average ratio of $C_1^+$ to $C_3^+$ was calculated to be 0.584 to correct the interference on $C_1^+$
from other organics. RIEs of ammonium, nitrate, sulfate and $r$BC were determined to
be 4.53, 1.10, 1.01, and 0.17, respectively, and RIE of organics used the default value
of 1.4 (Canagaratna et al., 2007). The size was calibrated by Polystyrene latex (PSL)
spheres (100–700 nm) (Duke Scientific Corp., Palo Alto, CA) before the measurement.
This study applied a collection efficiency (CE) of 0.5 for SP-AMS.
The co-located HR-AMS (DeCarlo et al., 2006) was used to measure all $PM_1$
including both $r$BCc and non-$r$BC particles, but it detected only non-refractory species
(NR-$PM_1$ species) as its 600 °C thermal heater is unable to vaporize $r$BC and other
refractory components. In addition, mass concentrations of gaseous pollutants, carbon





monoxide (CO), ozone (O₃), nitrogen dioxide (NO₂), and sulfur dioxide (SO₂) were
measured by the Thermo Scientific analyzers provided by SAES. Meteorological
parameters including air temperature (T), relative humidity (RH), wind speed (WS),
wind direction (WD) and precipitation, were obtained from Xujiahui Environmental
Monitoring Station of Shanghai (31°11'49.1424"N, 121°26'34.44" E)(~2400 m away
from the site). The concentrations of particle-phase vanadium (V) and nickel (Ni) that
were used to investigate ship influence were measured independently by an
Atmospheric heavy metal analyzer (XHAM-2000A, SAIL HERO., China).

**2.2 Data analysis**

The AMS data (both SP-AMS and HR-AMS) were analyzed using standard ToF-

AMS data analysis tool (Squirrel version 1.59D and Pika version 1.19D), based on Igor
Pro 6.37 (Wavemetrics, Lake Oswego, OR, USA). The mass concentrations and high
resolution mass spectra (HRMS) of $r$BC and coating species ($r$BC$_{CT}$) were calculated
from high-resolution (HR) fitting of V-mode data. Size distributions of $r$BCc
components were determined by the PToF data with unit mass resolution and were
scaled to their mass concentrations obtained above. In particular, size distribution of
$r$BC was scaled to that of $m/z$ 24 (C$_2^+$) (the scaling factor is $r$BC mass concentration to
that of calculated based on its size distribution), because $m/z$ 24 as a $r$BC fragment, has
least interference from other organic or inorganic species; such treatment was adopted
in earlier studies too (Collier et al., 2018;Wang et al., 2019;Wang et al., 2016).

The HR ion fitting of AMS data is able to distinguish various ions and isotopic

ions and calculate elemental ratios of organics such as oxygen-to-carbon (O/C),
hydrogen-to-carbon (H/C), nitrogen-to-carbon (N/C), and organic mass to organic
carbon (OM/OC) ratios, via the original Aiken-ambient (A-A) method (Aiken et al.,
2008) and the improved method (I-A) (Canagaratna et al., 2015b). Outcomes of both
methods correlated well. Average O/C, H/C, and OM/OC ratios from the I-A method
used in this work were 24.9%, 7.3%, and 5.6%, respectively, higher than those from the
A-A method.

Furthermore, we performed Positive matrix factorization (PMF) (Paatero and



Tapper, 1994) analysis on the HRMS of organics measured by the SP-AMS via the PMF
Evaluation Tool (Ulbrich et al., 2009). The PMF solutions were thoroughly evaluated
following the protocols documented in Zhang et al. (2011). Finally, a 6-factor solution
was chosen as the optimal one. The final result included four primary OA (POA) factors,
namely hydrocarbon-like species enriched OA (HOA-rich), $r$BC-enriched OA ($r$BC-
rich), biomass burning OA (BBOA), water-soluble hydrocarbon-like species enriched
OA (WS-HOA), and two secondary OA (SOA) factors including a less oxidized
oxygenated OA (LO-OOA$_{r\text{BC}}$), and a more oxidized oxygenated OA (MO-OOA$_{r\text{BC}}$)
(Key diagnostic plots in Fig. S2). The HR-AMS data were processed in a similar way,
and four factors of NR-PM$_1$ organics were resolved, including hydrocarbon-related OA
(HOA$_{\text{NR-PM1}}$), cooking OA (COA$_{\text{NR-PM1}}$), less oxidized oxygenated OA (LO-OOA$_{\text{NR-}}$
$_{\text{PM1}}$), and more oxidized oxygenated OA (MO-OOA$_{\text{NR-PM1}}$) (Mass spectra and time
series shown in Fig. S3).

**2.3 Calculation of size-resolved hygroscopicity of $r$BCc**
The hygroscopicity parameter $\kappa$ is a single parameter representing the
hygroscopicity of particles, calculated based on essentially the chemical composition
(Petters and Kreidenweis, 2007). SP-AMS measured size-resolved chemical
compositions of $r$BCc can thus lead to size-resolved hygroscopicity of $r$BCc ($\kappa_{r\text{BCc}}$)
(Wu et al., 2019;Hu et al., 2021). This study applied a similar method proposed by Wu
et al. (2019) and the critical parameters involved in calculation are detailed in Table S1.
The procedures are briefly described here: (1) obtain the matrice of size-resolved mass
concentrations of each $r$BCc component from SP-AMS analysis; (2) convert the size-
resolved concentration matrice of inorganic ions ( $SO_4^{2-}$ , $NO_3^-$ and $NH_4^+$ ) to
corresponding matrice of inorganic salts (NH$_4$NO$_3$, NH$_4$HSO$_4$, (NH$_4$)$_2$SO$_4$) using a
simplified solution of ion pairs (Gysel et al., 2007); (3) convert the size-resolved mass
matrice of all components (inorganic salts, OA and $r$BC) to those of size-resolved
volume fractions (Gysel et al., 2007;Chang et al., 2010;Wu et al., 2016); (4) combine
with previously reported hygroscopic parameters (Gysel et al., 2007;Chang et al.,
2010;Wu et al., 2016) to obtain the volumetric contributions of each component to the



hygroscopicity of $r$BCc; (5) use the Zdanovskii-Stokes-Robinson (ZSR) rule to derive
the size-resolved hygroscopicity of $r$BCc ($\kappa_{r\text{BCc}}$) (Topping et al., 2005a, b).

**3 Results and discussion**
**3.1 Overview of chemical characteristics of $r$BCc and NR-PM$_1$ particles**
Figure 1 presents the time series of meteorological parameters, concentrations of
gaseous pollutants (CO, NO$_2$, O$_3$, and SO$_2$), ship emission tracers (vanadium and
nickel), $r$BC and $r$BC$_{CT}$ species and their mass percentages to the total $r$BCc mass,
PMF-resolved OA factors and their corresponding fractional contributions. The
sampling period was featured by relatively moderate temperatures and very stagnant
conditions with average ($\pm 1\sigma$) temperature of 15.3 $\pm$ (2.9) °C and wind speed (WS) of
0.16 $\pm$ (0.29) m s$^{-1}$. Calm wind (<0.5 m s$^{-1}$) dominated most of the sampling days and
42% of sampling time was with near-zero wind, therefore overall influence of WS and
WD on surface mass loadings of $r$BCc was insignificant. Yet one should keep in mind
that WD can affect $r$BCc sources, and WS in higher altitude might be stronger therefore
long-range transport of air pollutants was still possible. The average concentrations of
CO, NO$_2$, O$_3$, SO$_2$, V, and Ni were determined to be 0.60 ppm, 29.20 ppt, 27.10 ppt,
1.27 ppt, 4.05 ng m$^{-3}$, and 3.06 ng m$^{-3}$, respectively.
The mass loadings of $r$BC and $r$BC$_{CT}$ ranged from 0.04 to 11.00 µg m$^{-3}$ and 0.37
to 30.47 µg m$^{-3}$ with campaign-mean values ($\pm$ 1σ) of 0.92 $\pm$ (0.81) µg m$^{-3}$ and 4.55 $\pm$
(4.40) µg m$^{-3}$. The coating materials accounted for 83.0% of the total $r$BCc mass, of
which organics was the most abundant species (2.54 $\pm$ 2.52 µg m$^{-3}$, 46.0%), followed
by nitrate (1.20 $\pm$ 1.30 µg m$^{-3}$, 20.0%), ammonium (0.44 $\pm$ 0.40 µg m$^{-3}$, 9.0%), sulfate
(0.30 $\pm$ 0.19 µg m$^{-3}$, 6.0%), and chloride (0.07 $\pm$ 0.05 µg m$^{-3}$, 1.0%). The mass ratio of
$r$BC$_{CT}$ to $r$BC ($R_{BC}$) ranged from 2.2 to 9.0, with an average of ~5.0 ($\pm$1.7). The average
$R_{BC}$ was higher than that in California ($R_{BC}$ = 2.3) (Collier et al., 2018) and in Shenzhen
($R_{BC}$ = 2.5)(Cao et al., 2022), lower than that in Tibetan Plateau ($R_{BC}$=7.7) (Wang et al.,
2017) and similar to that in Beijing ($R_{BC}$=5.0) (Wang et al., 2019), suggesting $r$BC was
relatively thickly coated throughout the campaign. Correlation between $r$BC and $r$BC$_{CT}$
was moderate (Pearson's $r^2$=0.58). Correlation coefficients ($r^2$) of chloride, nitrate,





sulfate and organics with $r$BC were 0.52, 0.75, 0.51 and 0.53, respectively, suggesting
variability of sources among different coating components.
Figure 2 compares the campaign-averaged diurnal patterns of $r$BC$_C$ and NR-PM$_1$
species, chemical compositions of $r$BCc and NR-PM$_1$, and mass ratios of the species
coated on $r$BC to those of NR-PM$_1$. We found that the diurnal variations of nitrate,
sulfate, ammonium, chloride were very similar ($r^2$>0.86) between the two particle
groups, while apparent difference was found for $r$BCc organics with a much obvious
morning rush hour peak. The results indicate that the formation processes of inorganic
salts coated on $r$BC were similar to those uncoated on $r$BC, but there were large
difference regarding sources/processes existed for organics. For $r$BC itself, the diurnal
cycle presented clearly a morning peak and an evening peak, likely relevant with rush
hour traffic emissions (CO showed similar pattern). On the contrary, $R_{BC}$ had a
minimum in the morning and dropped to a low level in later afternoon, probably due to
influence from traffic-emitted fresh and barely coated $r$BC particles (details in Section

3.2.1).

Distributions of species between $r$BCc and non-$r$BC particles were also different,
leading to different chemical compositions (Figs. 2g and 2h). Sulfate tended to
preferentially condense on non-$r$BC particles, as its mass contribution to total $r$BCc
mass was only 6.5%, while its contribution to total NR-PM$_1$ was 17.6%.
Apportionment of nitrate between $r$BCc and non-$r$BC particles was about even as it
both occupied ~26% of the total $r$BCc and NR-PM$_1$ masses. Organics occupied 55.9%
of $r$BCc mass, larger than it in NR-PM$_1$ (43.7%), due to that primary OA species
preferentially associated with $r$BC. Such result is similar to that observed in winter in
Beijing but contrary to the result that SOA was more abundant in $r$BCc in Shenzhen
(Cao et al., 2022).
On average, $r$BC$_C$ accounted for 19.1% of the total NR-PM$_1$ mass loading (21.61
± 15.80 μg m$^{-3}$)(Fig. 2i), comparable to that in Fontana, California (Lee et al., 2017).
The finding reveals that significant fractions of aerosol species were externally mixed
with $r$BC. Individually, sulfate captured by $r$BC only represented 7.4% of NR-PM$_1$
sulfate, similar to the earler results (Lee et al., 2017;Wang et al., 2020a;Cao et al.,



2022); mass fractions of $r$BCc nitrate (20.1%) and chloride (20.4%) in NR-PM$_1$ were
similar to the portion of total $r$BC$_C$ (19.1%), while the fraction of organics was higher
(26.1%). The relatively high ratio of organics was attributed to the fact that majority of
POA species were coated on $r$BC (average ratio of 72.7%), while $r$BC-related SOA was
21.8 % of the total. Note the $r$BCc POA here included all four POA factors, and COA$_{NR-}$
$_{PM1}$ did not coat on $r$BC thus was not included in the calculation.

**3.2 Distinctive sources of OA in $r$BCc and in NR-PM$_1$**
As shown previously, source apportionment results of OA in $r$BCc and NR-PM$_1$
were different. This section discusses in details the characteristics of OA sources in
$r$BCc and in bulk NR-PM$_1$.
**3.2.1 Source apportionment of OA in $r$BCc**
Figure 3 shows the HRMS and temporal variations of the six OA factors resolved
from PMF analysis of $r$BCc organics. Note the PMF analysis included $r$BC signals (i.e.,
$C_x^+$ ions) to aid identification of different factors, yet calculations of elemental ratios of
these OA factors did not include $C_x^+$ ions in order to explicitly explore the properties of
organic coating. The HRMS of HOA-rich and $r$BC-rich were similar to the OA
previously reported in urban environments near traffic emissions and/or in
gasoline/diesel vehicle exhaust (Massoli et al., 2012;Lee et al., 2015;Enroth et al.,
2016;Saarikoski et al., 2016;Willis et al., 2016;Lee et al., 2017), therefore both factors
were traffic-related. The HOA-rich mass spectrum was featured by intense hydrocarbon
ion series (i.e., $C_nH_{2n+1}^+$ and $C_nH_{2n-1}^+$ ions in Fig. 3c), and a lowest O/C ratio of 0.07.
Mass fraction of $r$BC signals (i.e., $C_n^+$ ions, such as $m/z$ 12, 24, 36, 48, and 60, etc.) in
HOA-rich was 8.1%. Mass spectrum of $r$BC-rich factor had remarkable contribution
from $r$BC (24.2%). Beside $C_n^+$ ions, the $r$BC-rich factor contained more oxygenated
organic fragments and presented a higher O/C ratio of 0.21 than that of HOA-rich,
similar to previous studies (Willis et al., 2016;Lee et al., 2017). This result is reasonable
as previous studies (Corbin et al., 2014;Malmborg et al., 2017) found that refractory
organics could generate oxygenated ion fragments (such as $CO^+$ and $CO_2^+$ derived from
oxygenated species on soot surface or inside soot nanostructure). HOA-rich factor



correlated very well with the common AMS tracer of vehicular OA, $C_4H_9^+$ ($r$=0.95, Fig.
3i), while $r$BC-rich factor did match the variation of $r$BC well ($r$=0.90, Fig. 3g). Since
diesel combustion often releases more $r$BC than that of gasoline, it is likely that the
$r$BC-rich factor is representative of diesel vehicle exhaust while HOA-rich factor
represents gasoline combustion emissions. This result demonstrates that laser-only SP-
AMS is capable of distinguishing diesel and gasoline burning particles which typically
cannot be separated by other AMS measurements. Further verification should be subject
of future work.

In this work, a multiple linear regression for the three-dimension size-resolved

mass spectra according to the method provided in Ulbrich et al. (2012) was used to
resolve the average size distributions of six OA factors. The diagnostic plots are shown
in Fig. 4. Overall, the lumped size distribution of six OA factor could reproduce well
that of total OA (except for a few size bins, most deviations are within 10%).
Correlation between measured and reconstructed OA of all size bins was very tight with
$r$ of 0.99 and a slope of 0.97, indicating the robustness of the regressed size distributions
of all OA factors. The results together with size distributions of $r$BCc components, and
corresponding mass fractional contributions of different components in all size bins are
illustrated in Fig. 5. The average HOA-rich size distribution peaked around 150 nm
($D_{va}$, vacuum aerodynamic diameter), generally matching with previously reported size
distribution of HOA (Sun et al., 2012;Ulbrich et al., 2012;Zhou et al., 2016).
Interestingly, size distribution of $r$BC-rich factor presented two modes, with one
peaking ~260 nm, and a more pronounced one peaking ~580 nm (Fig. 5a).

The BBOA was identified owing to its obviously higher signals of $C_2H_4O_2^+$ (2.03%)

and $C_3H_5O_2^+$ (1.62%) than those in other factors, as these two ions are well-known AMS
fragments of the biomass burning tracer, levoglucosan (Mohr et al., 2009;Cubison et
al., 2011). In addition, the time series of BBOA correlated particularly tightly with both
marker ions ($r$ of 0.86 and 0.80, respectively); it in fact also correlated well with K$^+$
($r$=0.79), another tracer of biomass burning emission. The O/C and H/C ratios of BBOA
were 0.12 and 1.78, and $C_n^+$ ions accounted for 9.1% of BBOA, all suggesting that the
factor was fresh and might be co-emitted with $r$BC. A relatively high N/C ratio (0.033)





was found for BBOA, which could be attributed to the large amounts of nitrogen-
containing organic species enriched in biomass burning OA (Laskin et al., 2009). In
addition, the size distribution of BBOA (Fig. 5a) (peak size ~500 nm) was similar to
that of biomass burning BC-containing particles obtained using single particle mass
spectrometry in Shanghai (Gong et al., 2016).

The PMF analysis deconvoluted a unique OA factor coated on $r$BC, namely WS-

HOA. Firstly, the WS-HOA mass spectrum had a series of hydrocarbon ion fragments
and its time series correlated well with them (e.g., $r$ of 0.90 and 0.92 for WS-HOA $vs$.
$C_4H_7^+$ and $C_4H_9^+$, respectively). Secondly, this factor correlated the best ($r$=0.57) with
aerosol liquid water content (ALWC) (Fig. 3j) compared with the other five OA factors
(all $r$<0.2). Thirdly, a previous study (Ye et al., 2017) that investigated specially the
water-soluble fraction of OA via PMF analysis, separated also a HOA factor that
contained significant nitrogen-containing organic fragments, with a highest N/C ratio
among all other factors, and correlated well those nitrogenated fragments. The WS-
HOA defined here showed similar characteristics with the highest N/C (0.037) among
all factors and tight correlations with nitrogen-containing fragments ($r$>0.80). At last,
although WS-HOA had a relatively high O/C (0.31) with remarkable contributions from
$C_2H_3O^+$ and $CO_2^+$, yet its correlations with these two ions were in fact not strong ($r$ of
0.46 and 0.44, respectively); and WS-HOA had the least fraction of $r$BC fragments
(0.9%) (note $r$BC is hydrophobic), even less than the two SOAs (Fig. 3d). Both results
suggest that this factor is a collection of water-soluble primary OA species. The peak
of WS-HOA size distribution was ~150 nm, close to that of HOA-rich (Figs. 5a).
Aqueous-phase processed SOA (aqSOA) were typically with very high O/C ratio (Xu
et al., 2019), and size distribution of aqSOA often presented a droplet mode, peaking in
relatively large sizes (such as 500~600 nm)(Gilardoni et al., 2016;Wang et al., 2021;Ge
et al., 2012). Therefore, the moderate O/C (0.31) and small mode size of WS-HOA
again manifest it was not from aqueous-phase reactions but more likely the water-
soluble fraction of POA.

The PMF analyses separated two SOA factors, LO-OOA$_{r\mathrm{BC}}$ and MO-OOA$_{r\mathrm{BC}}$.

Mass spectral features of the two SOAs were consistent with previous studies: The LO-
OOA$_{rBC}$ was rich in C$_x$H$_y$O$_1^+$ ions (38.7%) (such as C$_2$H$_3$O$^+$ at $m/z$ 43) but with less
contribution from C$_x$H$_y$O$_2^+$ ions (7.6%) and an overall moderate O/C (0.25), while MO-
OOA$_{rBC}$ had much more contribution from C$_x$H$_y$O$_2^+$ ion family (22.7%) (such as CO$_2^+$
at $m/z$ 44) and C$_x$H$_y$O$_1^+$ ions (44.7%), with a high O/C ratio (0.56). In addition, LO-
OOA$_{rBC}$ correlated better with nitrate ($r$=0.83) than it with sulfate ($r$=0.69), while the
correlation between MO-OOA$_{rBC}$ and sulfate ($r$=0.84) is better than it with nitrate
($r$=0.76). Size distributions of the two SOAs were also in accord with their secondary
behaviors, both accumulating at larger sizes (~450 nm for LO-OOA$_{rBC}$, and a bit larger
mode size of ~500 nm for MO-OOA$_{rBC}$), in agreement with previous observations (Sun
et al., 2012;Ulbrich et al., 2012;Zhou et al., 2016).

In total, traffic-related POA (sum of HOA-rich, $r$BC-rich and WS-HOA) was the

most abundant source of $r$BCc organics (39.1%); BBOA occupied ~18.4%; the
contributions of two SOAs were on par with each other (20.2% for LO-OOA$_{rBC}$, and
22.3% for MO-OOA$_{rBC}$) (Fig. 2g). Among traffic POA, gasoline derived HOA-rich
factor outweighed the diesel derived $r$BC-rich factor (11.4% $vs.$ 6.3% of the total $r$BCc).
Contributions of different factors varied greatly for different sizes of $r$BCc particles
(Fig. 5b). Small-sized particles were overwhelmingly dominated by traffic POA; SOA
contributions increased with increase of size, and dominated over POA for 300-800 nm
ones; contribution of BBOA was also relative larger for 300-800 nm than for other-
sized ones; the very large particles (800-1000 nm) were found to be affected mainly by
traffic POA in this work. Correspondingly, for the total $r$BCc particles, $r$BC cores
peaked at ~170 nm, while other secondary inorganic components, behaving like SOA
factors, all peaked at a big size (~550 nm) (Fig. 5c) and their mass percentages were
also large for large-sized particles (Fig. 5d).

Figure 6a further demonstrates the changes of mass fractions of each $r$BCc

component as a function of $R_{BC}$. $R_{BC}$ is a proxy of coating thickness. It was found that
the thinly coated $r$BCc ($R_{BC}$<3) were dominated (up to ~80%) by traffic-related POA.
With the increase of $R_{BC}$, contribution of secondary components increased gradually,
especially the two SOAs and nitrate (sulfate showed little changes across the whole $R_{BC}$
range); SOA and nitrate contributions reached 40.2% and 31.3% at $R_{BC}$ = 8, respectively.





Accordingly, the oxidation degree (OSc = 2*O/C-H/C) (Kroll et al., 2011) of coated
organics increased with $R_{BC}$. In Fig. 6b, we presented the mass contributions of OA
factors to the $r$BC mass at different $R_{BC}$ values. The $r$BC was as expected,
predominantly associated with POA (from 94% at $R_{BC}$<2 to 66% at $R_{BC}$ >8), similar to
those from Fontana (Lee et al., 2017). Contribution of $r$BC-rich factor decreased
obviously, and those of SOA factors increased with $R_{BC}$. The contributions of HOA-
rich, WS-HOA and BBOA factors changed little.

**3.2.2 Comparisons with NR-PM$_1$ organics**
As shown in Fig. S3, PMF analysis separated four OA factor for NR-PM$_1$ organics.
Two SOA factors (LO-OOA and MO-OOA) were resolved for both $r$BCc and NR-PM$_1$,
and their contributions to them were also close (Figs. 2g and 2h). Correlations of time
series between the two LO-OOA factors and between the two MO-OOA factors were
also tight ($r$ of 0.94 and 0.90, respectively), indicating similar formation processes for
each SOA. But, of course, the SOAs from $r$BCc and from NR-PM$_1$ were not entirely
the same, as later ones had higher O/C ratios (0.52 of LO-OOA$_{NR-PM1}$ and 0.62 of MO-
OOA$_{NR-PM1}$). On average, the portion of LO-OOA coated on $r$BC took up 21.6% mass
of LO-OOA in total NR-PM$_1$, and the portion was 26.0% for MO-OOA. This result
suggests that there were some but not big differences regarding the partitioning of LO-
OOA and MO-OOA onto $r$BCc and non-$r$BC particles.
Compared with SOAs, source apportionment results of POA were quite distinct.
Firstly, there was only one HOA factor resolved for NR-PM$_1$, while three HOA factors
were separated for $r$BCc. Note the $r$BC-rich and WS-HOA factors occupied merely
3.1% and 2.1% of NR-PM$_1$ OA mass, respectively, probably the cause that they were
not identified in NR-PM$_1$ OA. Nevertheless, mass loadings of the sum of HOA-rich,
$r$BC-rich and WS-HOA factors (termed as HOA$_{rBC}$) agreed quite well with that of
HOA$_{NR-PM1}$ ($r$=0.95) (Fig. 7a), verifying both source apportionment results. And,
HOA$_{rBC}$ took up 63.7% of HOA$_{NR-PM1}$, while previous studies reported that 81%
(Massoli et al., 2012) and 87 % (Massoli et al., 2015) of HOA were associated with
$r$BC. These results imply that HOA species in NR-PM$_1$ were largely internally mixed



with $r$BC affected by vehicular emissions. Secondly, COA$_{NR-PM1}$ was only identified in
NR-PM$_1$ OA. AMS-resolved COA$_{NR-PM1}$ is mainly from cooking oil and food itself,
therefore it negligibly internally mixes with $r$BC. This result is consistent with previous
observations (Lee et al., 2015;Willis et al., 2016;Lee et al., 2017;Collier et al., 2018).
At last, BBOA was identified in $r$BCc OA but not in NR-PM$_1$ OA. One plausible reason
was that the BBOA mass contribution was minor (equivalent to <5% of NR-PM$_1$ OA
mass); another speculation is that laser only SP-AMS can detect refractory species that
HR-AMS cannot, some of them might be originated from biomass burning (Wang et
al., 2020a).

Diurnal cycles of the POA and SOA factors from both PMF analyses are compared

in Figs. 7b and 7c. The diurnal pattern of stacked HOA$_{rBC}$ indeed agreed with that of
HOA$_{NR-PM1}$, both with two peaks in the morning and evening rush hours, and, the
patterns of $r$BC-rich, HOA-rich, and WS-HOA factors showed similar behaviors
individually (Fig. 7b). The diurnal variation of COA$_{NR-PM1}$ had pronounced peaks
during lunch and dinner times, and its percentage in NR-PM$_1$ OA reached 54% at night.
Diurnal patterns of two LO-OOA factors were somewhat different ($r$=0.35). LO-
OOA$_{rBC}$ has a major peak in the afternoon, while though LO-OOA$_{NR-PM1}$ concentration
rose in the afternoon too but peaked in early evening (~8 pm). The daily variations of
two MO-OOA factors were similar ($r$=0.83), both peaking at 16:00. The afternoon
increases of both SOAs indicate an important role of photochemical reactions, yet
differences in formation mechanisms, volatilities and partitioning behaviors of SOA
products could lead to diversities of their diurnal patterns and HRMS.

**3.3 Evolution and formation $r$BCc components**
**3.3.1 Behaviors of $r$BC cores**

Size distribution of $r$BC cores shown in Fig. 5c was relatively wide. Beside the

main peak at ~170 nm, it extended significantly into large sizes and had a small peak at
~550 nm, which was close to the peak of secondary components. With results shown in
Fig. 5a, we found that small $r$BC cores were often thinly coated, while thickly coated
$r$BCc particle were often highly aged and a portion of them also had large-sized $r$BC





cores. This result is likely owing to that oxidation of *r*BC-bounded organics and/or
condensation of secondary species onto *r*BC surface can induce restructuring of soot
aggregates to form compact and large cores (Chen et al., 2018;Chen et al., 2016). Such
phenomenon is in line with earlier studies (Liu et al., 2019;Gong et al., 2016). We
further show the image plot of size distributions of *r*BC at different $R_{BC}$ in Fig. S5a. It
can be found that the *r*BC mass in a large part concentrated in particles with $R_{BC}$ of 5-
8, and there was indeed a significant portion of *r*BC appearing in large size range (400-
800 nm) with very thick coating ($R_{BC}$ of 8-9).

**3.3.2 Formation of inorganic salts**

Sulfate and nitrate both peaked at a big size ~550 nm (Fig. 5c) and were mainly

associated with thickly coated *r*BCc ($R_{BC}$>6, Figs. S5d and S5e). To investigate the
impacts of photochemistry and aqueous/heterogeneous chemistry on the formation of
*r*BC$_{CT}$ species, we plotted the image plots of size distributions of nitrate, sulfate and
organics versus $O_X$ ($O_3 + NO_2$) and relative humidity (RH) in Fig. 8. Here $O_X$ is used
as a proxy of photochemical activity (Xu et al., 2017), and RH is an indicator of aqueous
reactions (Wu et al., 2018).  Nitrate significantly concentrated in 65-100 ppb $O_X$ range
but there was a weak accumulation in low $O_X$ as well (Fig. 8a), while in Fig. 8d, nitrate
had a prominent hotspot in RH>85%. Generally, both strong photochemical activity
and high RH could promote nitrate formation. For sulfate, although the distribution was
scattered due to low level of mass loadings, high Ox level seemed to favor sulfate
formation (Fig. 8b); sulfate was scattered in the whole RH range and there were some
enhancements at high RH (>80%) but was much less clear-cut (Fig. 8e). Therefore
aqueous-phase production of sulfate was not important in this campaign.

We further calculated the sulfur oxidation ratio (SOR) and nitrogen oxidation ratio

(NOR) (Xu et al., 2014), and plotted their variations against $O_X$ and RH in Figs. 9a and
9e, respectively. The NOR rose substantially at $O_X$>60 ppb but showed no increase at
$O_X$ < 60 ppb, while it increased continuously with RH. Mass ratio of nitrate to *r*BC
stayed at a high level during nighttime when RH was high as well (overall diurnal trend
of $NO_3^-$/*r*BC was similar to that of RH, see Figs. S6a and S6d). This result indicates a


likely more important role of nocturnal nitrate formation ($N_2O_5$ hydrolysis) (Pathak et
al., 2011) (Sun et al., 2011) than photochemical production of nitrate during this
campaign; moreover, low temperature and high RH favor nitrate partitioning into the
particle phase during nighttime too (Gao et al., 2011). For sulfate, The SOR increased
with $O_X$ while it increased with RH under relatively dry conditions (<60%) but
decreased with RH when RH>60%. This result, on the other hand, highlights that
photochemical production is more important than aqueous/heterogenous formation for
sulfate. In addition, mass ratios of sulfate to $r$BC were enhanced remarkably during
daytime and peaked in the afternoon (Fig. S6e), supporting the key role of
photochemical formation of sulfate. Sulfate precursor $SO_2$ was at a high level during
daytime too. The main formation pathway of sulfate is strikingly different from that
observed in winter Nanjing (Wu et al., 2018), suggesting significant seasonal variability
of sulfate formation even in the same region.

**3.3.3 Evolution of organics**
Organics had a broad average size distribution (Fig. 5c), but unlike $r$BC, its main
peak appeared at 500~600 nm. Figure S5b shows that the majority of organics
partitioned in $r$BCc with $R_{BC}$ of 5.0-9.0 and wide size coverage (300-800 nm).
Regarding its dependences on $O_X$ and RH, it mainly accumulated at $O_X$>70 ppb (Fig.
8c) and very high RH (~90%) (Fig. 8f). Consistently, O/C ratio and OSc both peaked
in the afternoon (Fig. S6b), all suggesting a critical role of photochemistry in affecting
the behavior of organics.
Figure 9 illustrates the mass ratios of each OA factor to $r$BC varying with $O_X$ and
RH. Mass ratios of all four POA factors generally presented decreasing trends (despite
some fluctuations) against $O_X$ (Fig. 9b), and the total $POA_{rBC}$ showed evident decrease
with increase of $O_X$ (Fig. 9d). Instead, both $LO\text{-}OOA_{rBC}$ and $MO\text{-}OOA_{rBC}$, as well as
their sum ($SOA_{rBC}$) increased continuously with $O_X$ (Figs. 9c and 9d). This result proves
that photochemical oxidation contributed significantly to both $LO\text{-}OOA_{rBC}$ and MO-
$OOA_{rBC}$ formations. Comparatively, decreases of $POA_{rBC}$ perhaps point to its reaction
loss upon photochemical oxidation. With regard to RH, besides WS-HOA, the other



three POA$_{rBC}$ factors showed almost no dependences on RH (Fig. 9f); note the increase
of WS-HOA with RH did not indicate the aqueous production of WS-HOA (see
discussion in Section 3.2.1), but a result of enhanced dissolution with increase of
moisture. Overall small increase of POA$_{rBC}$ (Fig. 9h) with RH then mainly attributed to
WS-HOA. Contrary to the trends with O$_X$, mass ratios of two SOA factors as well as
the total SOA to $r$BC went down with increase of RH (Figs. 9g and 9h), indicating a
trivial role of aqueous/heterogenous oxidation for the SOA coated on $r$BC observed
during this campaign.

The aging of OA is generally characterized by the increase of O/C and decrease of

H/C (Ng et al., 2011;Zhao et al., 2019).The different aging pathways of OA follow
different slopes in the Van Krevelen (VK) diagram (Heald et al., 2010). For example,
addition of only one oxygen atom to the carbon skeleton results in a slope equal to 0,
while replacement of the hydrogen atom with a carboxylic acid group (−COOH) results
in a slope of −1 without fragmentation (C-C bond breaking), and −0.5 with
fragmentation (Heald et al., 2010;Ng et al., 2011;Zhao et al., 2019). As presented in Fig.
10a, fitting of all OA data yielded a slope of −0.96, very close to −1, suggesting that
OA ageing process resembled the hydrogen substitution with a –COOH group
(carboxylation). Interestingly, the four OA factors (HOA-rich, WS-HOA, LO-OOA$_{rBC}$
and MO-OOA$_{rBC}$) aligned almost in a straight line with a slope of −0.77 (Fig. 10a), also
close to −1. Figure S7 further reveals that there was a strong anti-correlation between
mass fractions of sum of HOA-rich and WS-HOA and sum of LO-OOA$_{rBC}$ and MO-
OOA$_{rBC}$ ($r$=−0.97); the slope of fitted line was −0.86. All these results suggest that OA
evolution may contain a channel of photochemical transformations from HOA-rich and
WS-HOA to LO-OOA$_{rBC}$ and then to MO-OOA$_{rBC}$. This result is also in line with the
observed decrease of POA$_{rBC}$ and increase of SOA$_{rBC}$ against O$_X$.

CHO$^+$, CHO$_2^+$ and C$_2$H$_2$O$_2^+$ are the AMS tracer ions for carbonyl, carboxylic acid

and glyoxal, respectively (Wang et al., 2020b;Canagaratna et al., 2015a;Yu et al., 2014).
Mass fractions of these three fragment ions presented decreasing trends (or no clear
trends) against RH (Fig. S8), suggesting again that aqueous processing is not an
important pathway in OA evolution during this campaign. Conversely, fractional





contributions of these three ions presented increasing trends versus $O_X$, supporting the
dominance of photochemical oxidation pathway (Figs. 10b-d). Figures 10e-g show the
scatter plots of H/C versus O/C at different $O_X$ concentrations. The regressed slope was
−1.03 for low $O_X$ (<60 ppb) conditions (Fig. 10e), indicating that the OA aging at low
$O_X$ level is mainly analogue to the carboxylation process. This result corresponds
precisely to the fact that mass fractions of $CHO_2^+$ and $C_2H_2O_2^+$ increased gradually with
$O_X$ at low $O_X$, whereas that of $CHO^+$ remained essentially unchanged (Figs. 10e-g). The
fitted slope was −1.14 for moderate $O_X$ conditions (60-80 ppb), and it changed to −0.43
for high $O_X$ level (>80 ppb) but correlation became weaker ($r$=−0.57). This result
implies that the OA evolution under high $O_X$ conditions might include oxidations by
the additions of alcohols, peroxides and carboxylation. In all, the evolution of $r$BCc OA
in Shanghai during this campaign is governed by photochemistry rather than aqueous
chemistry, but with different oxidation pathways at different $O_X$ levels.

**3.3.4 Coating time of secondary species onto $r$BC**

Although the $r$BCc organics was dominated by primary sources (Fig. 2g), the

diurnal variations of OSc, O/C and H/C of the total organics, were controlled
predominantly by the two SOA factors. Correlations between the diurnal cycles of MO-
OOA$_{rBC}$/$r$BC and OSc were extremely well ($r$=0.97 with $OS_C$ and $r$=0.98 with O/C),
and those of LO-OOA/$r$BC were also tight ($r$=0.91 with $OS_C$ and $r$=0.92 with O/C).
The correlations with LO-OOA$_{rBC}$ were a bit weaker than those of MO-OOA$_{rBC}$,
indicating that MO-OOA$_{rBC}$ was probably the final products and was more important
in governing the overall oxidation level of organic coating. Figure 11a depicts the
diurnal variations of SOA$_{rBC}$/$r$BC and POA$_{rBC}$/$r$BC. Diurnal variations of POA$_{rBC}$/$r$BC
and $r$BC were overall similar, while the daily pattern of SOA$_{rBC}$/$r$BC was almost
opposite to that of $r$BC. This result likely indicates that most POA$_{rBC}$ species were co-
emitted and coated on $r$BC cores originally, therefore the coating process during $r$BCc
lifecycle was mainly relevant with SOA species rather than POA species. This process
is mainly through photochemical reactions, including in-situ oxidation of originally
coated POA species (for example, oxidation of HOA-rich plus WS-HOA to LO-



OOA$_{rBC}$, then to MO-OOA$_{rBC}$), and partitioning of secondary species formed in gas-
phase reactions onto $r$BC surface. In addition, sulfate and nitrate were both secondarily
formed, but sulfate was dominated by photochemical production while nitrate was
governed by nocturnal heterogenous formation (as discussed in Section 3.3.2); different
diurnal patterns of them (Fig. S6) point to different coating processes too.

In this regard, we hereby propose a concept of average coating time (ACT), which

is used to roughly estimate the timescales required for secondary components coated
onto $r$BC. The specific method is listed as follows:

1. Move forward the diurnal variation of SA$_{rBC}$/$r$BC (SA represents a secondary

aerosol species) for $n$ hours to get a new SA$_{rBC}$/$r$BC diurnal pattern, labelled as
"SA$_{rBC}$/$r$BC($r$-$n$h-ahead)". Here, the $r$ value is the linear correlation coefficient between
the new SA$_{rBC}$/$r$BC diurnal pattern with that of $r$BC.

2. Choose the best correlation coefficient ($r$-$n$h-ahead), and $n$h corresponds to the

ACT for this secondary component.

Diurnal patterns of LO-OOA$_{rBC}$/$r$BC and MO-OOA$_{rBC}$/$r$BC are depicted in Fig.

11b. They were both opposite to the trend of $r$BC, and they were similar to each other,
except that MO-OOA$_{rBC}$/$r$BC peaked hours later in the afternoon, signifying that the
MO-OOA$_{rBC}$ needs longer time to coated on $r$BC than LO-OOA$_{rBC}$ does, consistent
with the fact that MO-OOA$_{rBC}$ was "more aged". Correspondingly, we obtained an ACT
of 7 hours for MO-OOA$_{rBC}$ (*0.35-7*h-ahead) and 5 hours for LO-OOA $_{rBC}$ (*0.57-5*h-
ahead) (Fig. 11c) using the method described above. The ACT of sulfate (*0.65-7*h-ahead)
and nitrate (*0.30-19*h-ahead) were 7 and 19 hours, respectively (Fig. 11d). The results
suggest that the $r$BC emitted mainly in the morning rush hours requires a few hours to
be adequately coated by LO-OOA$_{rBC}$, MO-OOA$_{rBC}$ and sulfate, as these three species
are mainly produced in the afternoon by photochemical reactions; while photochemical
production of nitrate is insignificant, thus $r$BC was coated by nitrate until later night
when nitrate was formed efficiently by heterogenous $N_2O_5$ hydrolysis. Note the best $r$
values were not high  (for example, 0.35 for MO-OOA$_{rBC}$ and 0.30 for nitrate),  yet the
adjusted trends of all secondary components (Figs. 11c and 11d) matched that of $r$BC
quite well during 3:00~12:00 ($r$ of 0.90, 0.91, 0.84 and 0.84 for MO-OOA$_{rBC}$, LO-





OOA$_{r\mathrm{BC}}$, sulfate and nitrate, respectively), likely meaning that daytime variations of
two SOAs and sulfate were indeed controlled by the coating process while governing
factors of their nighttime variations might be complex, and *vice versa* for nitrate.

**3.4 Size-resolved hygroscopicity of *r*BCc**
By using the method in Section 2.3, we calculated the size-resolved hygroscopicity
parameters for the total *r*BCc ($\kappa_{r\mathrm{BCc}}$) and the coatings materials ($\kappa_{\mathrm{CT}}$) across the whole
campaign. The image plots are illustrated in Figs. 12a and b. Generally, large $\kappa_{r\mathrm{BCc}}$ and
$\kappa_{\mathrm{CT}}$ values occurred at big particle sizes, and this result was overall similar to that in
Nanjing during winter (Wu et al., 2019). We further illustrate the size-resolved $\kappa_{r\mathrm{BCc}}$ as
a function of $R_{\mathrm{BC}}$ in Fig. 12c. The figure shows that $\kappa_{r\mathrm{BCc}}$ overall became larger with
increasing particle size regardless of the coating thickness. However, there were some
(though not significant) relatively high $\kappa_{r\mathrm{BCc}}$ values in the range of 80-150 nm (bottom
left and bottom right in Fig. 12c).
In Figs. 13a and b, we further determined the average size-resolved $\kappa_{r\mathrm{BCc}}$ and $\kappa_{\mathrm{CT}}$
as a function of coated diameter ($D_{r\mathrm{BCc}}$). Both $\kappa_{r\mathrm{BCc}}$ and $\kappa_{\mathrm{CT}}$ values were relatively high
at $D_{r\mathrm{BCc}}$<100 nm and presented slight decreases from 100 to 150 nm. This is distinctive
from those observed in Nanjing, where $\kappa_{\mathrm{CT}}$ increased with $D_{r\mathrm{BCc}}$ from 50 nm (Wu et al.,
2019). From Figs. 5b and 5d, it can be seen the *r*BCc with $D_{r\mathrm{BCc}}$<150 nm was dominated
by a portion of ammonium and sulfate (8-10%) and organics (~60%), of which organics
was dominated by WS-HOA. Such composition explains the relatively high
hygroscopicity at $D_{r\mathrm{BCc}}$<150 nm as both ammonium sulfate and WS-HOA are
hydrophilic; a slight decrease of the hygroscopicity from 100 to 150 nm $D_{r\mathrm{BCc}}$ was also
a response of decreased mass contributions of ammonium sulfate and WS-HOA (and
increase of hydrophobic HOA-rich contribution).
Figures 13a and b provide the fitted exponential functions for the mean $\kappa_{r\mathrm{BCc}}$ and
$\kappa_{\mathrm{CT}}$ with $D_{r\mathrm{BCc}}$. The equations are: $\kappa_{r\mathrm{BCc}}(x)= 0.29-0.14 \times \exp(-0.006 \times x)$ and $\kappa_{\mathrm{CT}}(x) =$
$0.35 - 0.09 \times \exp(-0.003 \times x)$ ($x$ is $D_{r\mathrm{BCc}}$, 150<$x$<1000 nm). Here, 0.29 and 0.35 are
the upper limits of $\kappa_{r\mathrm{BCc}}$ and $\kappa_{\mathrm{CT}}$, higher than those reported in Nanjing (0.28 and 0.30
for $\kappa_{r\mathrm{BCc}}$ and $\kappa_{\mathrm{CT}}$); yet the increasing rates of $\kappa_{r\mathrm{BCc}}$ and $\kappa_{\mathrm{CT}}$ with $D_{r\mathrm{BCc}}$ are 0.14 and 0.09,



respectively, which are much lower than those from Nanjing (0.35 and 0.27 for $\kappa_{r\text{BCc}}$
and $\kappa_{\text{CT}}$)(Wu et al., 2019). Smaller increased contributions of hydrophilic secondary
species from 150 to 1000 nm in Shanghai than those from 100 to 1000 nm in Nanjing
are likely the cause of smaller increasing rates of hygroscopicity parameters. In addition,
it should be noted that the hygroscopicity is not only determined by the bulk
composition, but also affected by the phase state of particles. For instance, a recent
study reveals that the hygroscopic growth of phase-separated particles (with ammonium
sulfate as cores) can be reduced by the secondary organic shells and is dependent on
the thickness of organic coating (Li et al., 2021).

The critical supersaturation ($SS_C$) for a selected dry diameter ($D_{r\text{BCc}}$ measured by

SP-AMS) of a $r$BCc particle with a hygroscopicity parameter $\kappa_{r\text{BCc}}$, can be calculated
by the "$\kappa$-Kohler theory" equation (Petters and Kreidenweis, 2007). Based on the size-
resolved $\kappa_{r\text{BCc}}$, the CCN activation diameter ($D_{50}$) of particles at a given critical $SS_C$ can
be calculated (Wu et al., 2019). Then, by combining the measured $r$BCc number size
distribution and the $D_{50}$ value, activation fraction ($f_{\text{AC}}$) of $r$BCc number population (i.e.,
the fraction greater than the $D_{50}$) can be obtained. Figure 13c shows the $SS_C$ as a
function of $D_{r\text{BCc}}$ for the entire sampling period to obtain the $D_{50}$ at a specific $SS_C$. The
$D_{50}$ values of the $r$BCc particles were determined to be $166 \pm 16$ nm and $110 \pm 5$ nm
for $SS_C$ of 0.1% and 0.2%, respectively. The two $D_{50}$ values are both smaller than those
determined for $r$BCc particle in Nanjing (Wu et al., 2019), likely owing to the presence
of WS-HOA in Shanghai. Figure 13d shows the $f_{\text{AC}}$ at $SS_C$ of 0.1% ($D_{50}$ of 166 nm) was
$16 \pm 3\%$, and the $f_{\text{AC}}$ at $SS_C$ of 0.2% ($D_{50}$ of 110 nm) was $59 \pm 4\%$.

**3.5 A case study influenced by ship emissions**
**3.5.1 Potential source areas of $r$BCc**

To explore the potential geographic origins of $r$BCc at the receptor site, the hybrid

single-particle Lagrangian integrated trajectory (HYSPLIT) model (version 4.9) was
applied here. Figure 14a shows that the backward trajectories were classified into four
air mass clusters, including one long-range transport from northern sea (Cluster1), one
long-range transport from northeastern sea (Cluster2), a local one from eastern ports





(Cluster3), and one from northwestern inland region (Cluster4). The four clusters
occupied 23.8%, 33.8%, 37.3%, and 5.0% of the total trajectories, respectively. It is
clear that the sampling period was influenced by offshore air masses in most of the time
(95%). Cluster3 had the highest mean $r$BCc concentrations ($13.2 \pm 10.9$ μg m$^{-3}$) while
the mean concentrations of the other three clusters were apparently lower (4.3~5.2 μg
m$^{-3}$). This result is plausible as Cluster3 was the shortest in length therefore was least
diluted compared with other three clusters. Average chemical compositions of the $r$BCc
from four clusters (Fig. 14b) showed some differences as well: Cluster1 and Cluster2
had higher MO-OOA$_{rBC}$ contributions, possibly owing to the interceptions of more
aged SOA species during the long-range transports; While Cluster4 had less MO-
OOA$_{rBC}$ but a bit more POA$_{rBC}$ contributions, likely attributing to more primary species
emitted in inland regions (similarly, a higher fraction of nitrate was likely because of
enhanced NO$_2$ emissions).

**3.5.2 A typical case of $r$BCc affected by ship emissions**
Ship emissions are found to have an important impact on the air quality of
Shanghai and the Yangtze River Delta (Zhao et al., 2013;Fan et al., 2016;Liu et al.,
2017b;Chen et al., 2019). The ship engines usually burn heavy fuel oil (HFO), and
vanadium (V) and nickel (Ni) can be adopted as reliable tracers for the ship-emitted
particles (Ault et al., 2009;Moldanová et al., 2009;Ault et al., 2010). The long-term
variation of Ni/V ratio in ship-emitted particles in Shanghai has been recently reported
(Yu et al., 2021). The main conclusion is that Ni/V ratio was close to 0.4 in 2018, while
it became to be greater than 2.0 in 2020. Our measurement was carried out in 2018, and
we therefore chose a period from November 3 to 5 as a typical case affected by ship
emissions (SEP period), since the average Ni/V ratio was ~0.50 and high concentrations
of both Ni and V were found. Figure S9 shows the concentration-weighted trajectories
(CWT) of ship emission tracers (V, Ni), $r$BC and $r$BC-rich factor during SEP, displaying
that these components were mainly from sea. During SEP, the correlation coefficient ($r$)
between V(Ni) and $r$BC was 0.69 (0.74), indicating the SEP was indeed impacted by
ship emissions.





Figure 15 displays the chemical characteristics of $r$BCc and NR-PM$_1$ components
(especially the OA factors), V and Ni, gaseous pollutants and the meteorological
parameters during SEP. As a comparison, we also selected a period with no ship
emissions with the same time span as SEP (from 0:00 on November 10 to 0:00 on
November 12, termed as non-SEP), and a similar plot during non-SEP is presented in
Fig. S10. During SEP, the wind was mainly from east, and the average wind speed was
~0.5 m s$^{-1}$ (Fig. 15a); Wusong, Luojing and Waigaoqiao ports located northeast of the
sampling site (Fig. S1). Instead, the wind was mainly from north during non-SEP (Fig.
S10a). During SEP, the average mass concentrations of V and Ni were 5.8 ng m$^{-3}$ and
2.9 ng m$^{-3}$, respectively, while those during non-SEP were only 2.9 ng m$^{-3}$ and 2.6 ng
m$^{-3}$. The average mass ratio of V/Ni during SEP was 2.0 in agreement with those
reported in ship-influenced PM$_1$ (Mazzei et al., 2008;Mar et al., 2009), and within the
range of 1.9 to 3.5 for domestic HFO (Zhao et al., 2013), while the average ratio of
V/Ni (1.1) during non-SEP was outside the range. Moreover, the major air pollutants
emitted from ships include nitrogen oxides (NO$_x$), sulfur oxides (SO$_2$), carbon
monoxide (CO), hydrocarbons and primary/secondary particles (Becagli et al.,
2017;Wu et al., 2021). As demonstrated in Fig. 15d and Fig. S10d, SO$_2$ concentration
was overall higher during SEP (10:00-16:00 on November 11 not included); mass
loading of NO$_2$ was higher during the rush hours of non-SEP, but was higher at night
(no traffic) during SEP. Mass proportion of sulfate in NR-PM$_1$ during SEP was also
higher than that during non-SEP (21.0% *vs.* 18.7%), but *vice versa* for nitrate (18.8%
during SEP *vs.* 25.4% during non-SEP).
We further investigated the dependences of $r$BCc and NR-PM$_1$ species on V during
SEP and non-SEP, as shown in Fig. S11. Here we only used V since Ni level might be
influenced by other emission sources, such as refining industry (Jang et al., 2007;Kim
et al., 2014) in urban Shanghai, and during non-SEP, Ni still presented a good
correlation with $r$BC ($r$=0.80). During SEP, V concentrations (most of them >4 ng m$^{-3}$)
overall positively correlated with both sulfate and nitrate (except for $r$BCc sulfate) (Fig.
S11a). Considering that V concentration was independently measured for all fine
particles, a better correlation with total NR-PM$_1$ sulfate than it with $r$BCc sulfate is





reasonable. Similarly, V-rich particles had positive correlations with traffic-related OA
and LO-OOA no matter in $r$BCc or in total NR-PM$_1$ (Fig. S11b). Conversely, during
non-SEP, particles with low-V content (most of them <4 ng m$^{-3}$) had no clear links with
sulfate, nitrate, POA and SOA species (even anti-correlations for V>2.5 ng m$^{-3}$) (Figs.
S11c and S11d; a detailed comparison of the correlation coefficients of V with OA
factors of $r$BCc and NR-PM$_1$ organics are provided in Table S2). The results above
dodemonstrate that during SEP, chemical properties of the particles (both fresh and aged
ones) were clearly linked with ship influences.
Previous studies (Ault et al., 2009;Ault et al., 2010;Liu et al., 2017b) have shown
that the fresh ship-emitted V-rich particles are typically accompanied by high sulfate
contribution, while those aged V-containing ship particles are on the other hand with
relatively high nitrate contribution. In order to further probe chemical characteristics
and evolution processes of particles influenced by ship emissions, we divided SEP into
three episodes (i.e., EP1-EP3) (marked in Fig. 15). During EP1, $r$BC content was
highest (Fig. 15k) and coating was thinnest (Fig. S12i) indicating the particles were
relatively fresh, nevertheless the SOA contents were not low (Figs.15j and 15l), OS$_C$
was moderate (Fig. S12j), sulfate portions in NR-PM$_1$ and $r$BCc were both the highest
(26.5%), and nitrate portion was the lowest (9.8%) among three episodes (Figs.15i and
15k). Such composition is not common in other cases, demonstrating it was a specific
period impacted by fresh ship emissions. EP2 had the highest mass loadings of V, gas
pollutants as well as the lowest planetary boundary layer (PBL) height (~200 meters)
(Fig. S12). Mass contributions of nitrate increased and sulfate decreased, and $r$BC
content decreased from those during EP1, signifying that the particle population likely
contained some aged ship-emitted particles. Of course, particle composition during EP2
was also influenced by the formation mechanisms of secondary species: EP2 was
mostly during nighttime, therefore photochemical formation of sulfate and SOA were
weak (contributions were low as shown in Figs. 15i-l), while nitrate formation was
enhanced due to the nocturnal process. During EP3, $r$BC was lowest, sulfate and V were
moderate, nitrate and SOA were highest in both $r$BCc and NR-PM$_1$, OS$_C$ and $R_{BC}$ were
highest in $r$BCc among the three episodes (Figs. 15i-l and Figs. S12i-k), therefore it





was also a period with influence from aged ship-emitted particles; the difference from
EP2 is that photochemically formed sulfate and SOA were still significant as EP3 was
in the later afternoon and early evening, though heterogeneously formed nitrate played
a non-negligible role too (see the increase of RH, increase of nitrate and decrease of $O_3$
concentrations from the beginning of EP3 in Fig. 15).

**4. Conclusions and implications**
We conducted a field measurement during November of 2018 in urban Shanghai,
China, focusing on the elucidation of physical and chemical properties of the ambient
particles containing $r$BC cores ($r$BCc) by using a laser-only SP-AMS together with a
HR-AMS. The campaign-average $r$BCc was 4.6 μg m$^{-3}$, occupying ~19.1% mass of the
total NR-PM$_1$. The average mass ratio of coating to $r$BC cores ($R_{BC}$) was ~5.0,
indicating an overall thick coating, compared with the $r$BCc near combustion source.
Sulfate was found to preferentially condense on non-$r$BC particles thus led to a low
fraction of $r$BCc sulfate to that in NR-PM$_1$ (7.4%), while distribution of nitrate between
$r$BCc and non-rBC particles showed no obvious difference. PMF analysis on $r$BCc and
NR-PM$_1$ OA reveals that cooking-related organics were externally mixed with $r$BC,
and a small portion of organics from biomass burning was only present in $r$BCc; the
traffic-related OA species, however, was in a large part internally mixed with $r$BC.
A regression algorithm was applied to deconvolute the size distributions of
individual $r$BCc OA factors, and results show that small $r$BCc particles were
predominantly generated from traffic, and such particles could grow bigger because of
condensation of secondary inorganic and organic components, resulting in thick coating.
Investigation on diurnal patterns of the $r$BCc species reveals that sulfate and two SOA
factors (LO-OOA$_{rBC}$ and MO-OOA$_{rBC}$) were generated mainly through daytime
photochemical oxidation; nitrate, on the other hand, was governed mainly by the
nocturnal $N_2O_5$ hydrolysis. Partial SOA was found to be produced from in-situ
photochemical conversion from traffic-related POA. An average coating time (ACT)
was proposed to roughly estimate the timescales for the secondary species to coat on
$r$BC, and the ACT of sulfate, LO-OOA$_{rBC}$, MO-OOA$_{rBC}$ and nitrate were approximately



5, 5, 7 and 19 hours, respectively.
Moreover, the size-resolved hygroscopicity parameters of $r$BCc particles ($\kappa_{r\text{BCc}}$)
and the coating material ($\kappa_{\text{CT}}$) were obtained based on the elucidated composition of
$r$BCc particles. The fitted equations are: $\kappa_{r\text{BCc}}(x)= 0.29-0.14 \times \exp(-0.006 \times x)$ and
$\kappa_{\text{CT}}(x) = 0.35 - 0.09 \times \exp(-0.003 \times x)$ ($x$ is $D_{r\text{BCc}}$, $150<x<1000$ nm). The minimums
of both $\kappa_{r\text{BCc}}$ and $\kappa_{\text{CT}}$ were at ~150 nm due to the abundances of hydrophobic $r$BC cores
and traffic-related HOA at this size. Under critical supersaturations ($SS_{\text{C}}$) of 0.1% and
0.2%, the $D_{50}$ values were estimated to be $166 \pm 16$ and $110 \pm 5$ nm, and the activated
number fractions of $r$BCc particles were $16 \pm 3\%$ and $59 \pm 4\%$, respectively. Finally, a
typical case with influence from ship emissions was investigated. During this period,
the $r$BCc particles were enriched in V (on average 5.8 ng m$^{-3}$), with a V/Ni ratio of 2.0,
and various secondary formation processes affect the ship-emitted particles at different
times of the day.
In summary, the findings from this comprehensive study on $r$BCc provide rich
information regarding the various primary sources and secondary formation pathways
of species coated on $r$BC, as well as the features of distributions of those species
between $r$BC and non-rBC particles. In particular, different types of $r$BCc particles
from diesel and gasoline vehicle emissions were resolved and elucidation of their
properties are useful to their future effective control. Understanding of the formation
processes and coating timescales of secondary components is helpful to understand the
impact of ambient BC particles too. At last, the parameterized relationship of
hygroscopicity with size distribution is useful for modelling the climate effect of $r$BC
(alternation of cloud properties).

*Data availability.* The data in this study are available from the authors upon request
(caxinra@163.com).

*Supplement.* The supplement related to this article is available online at: XXX

*Author contributions.* SJC, DDH, YZW, JFW, FZS, and XLG conducted the field



measurement. SJC, DDH, YZW, JFW, and JKX analyzed the data. YJZ, HLW, CH and
HL reviewed the paper and provide useful suggestions. SJC and XLG wrote the paper.

*Competing interests.* The contact author has declared that neither they nor their co-
authors have any competing interests.

*Disclaimer.* Publisher's note: Copernicus Publications remains neutral with regard to
jurisdictional claims in published maps and institutional affiliations.

*Acknowledgements.* We sincerely thank SAES to provide data of gaseous pollutants and
particulate V and Ni, and the logistic help during the campaign.

*Financial support.* This work has been supported by the National Natural Science
Foundation of China (42021004 and 21976093).

*Review statement.* This paper was XXX.

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

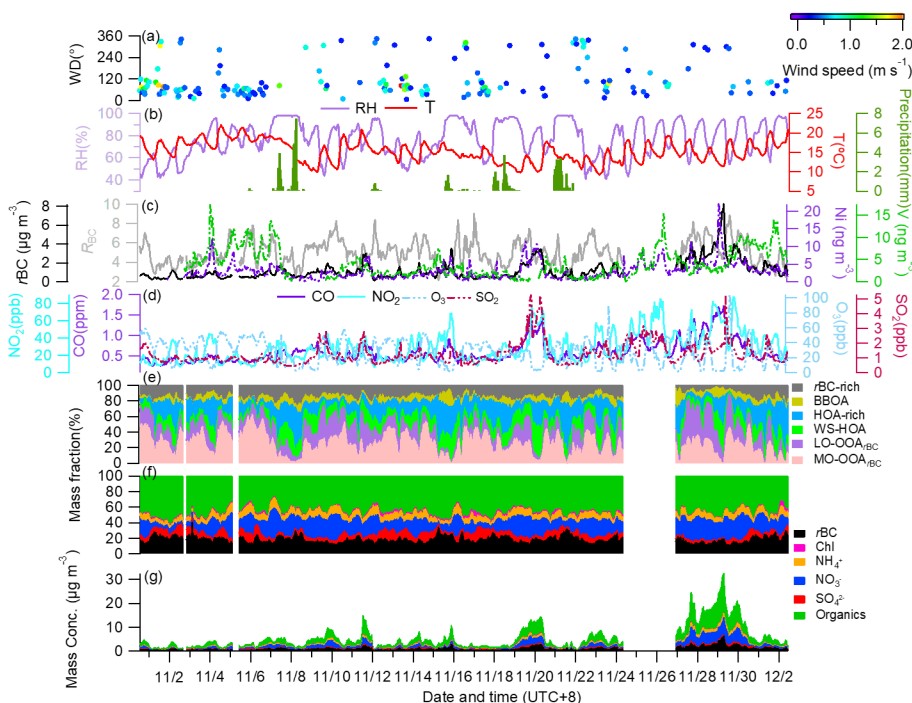

Figure 1. Time series of (a) wind direction (WD) colored by wind speed (WS), (b) air temperature (T), relative humidity (RH) and precipitation, (c) mass concentrations of $r$BC, Ni, V, and $R_{BC}$ (mass ratio of all coating species to $r$BC), (d) mass concentrations of gas pollutants of CO, NO$_2$, O$_3$ and SO$_2$, (e) mass fractions (%) of different OA factors to the total $r$BCc OA, (f) mass fractions (%) of different components to the total $r$BCc mass, and (g) mass concentrations of stacked $r$BCc components.



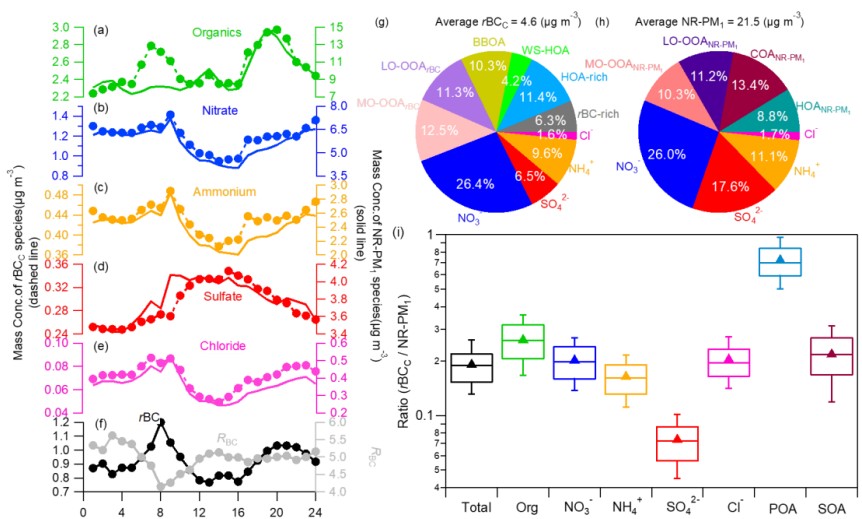

Figure 2. Diurnal cycles of mass concentrations of (a–e) $r$BCc and NR-PM$_1$ species (organics, nitrate, ammonium, sulfate, and chloride), and (f) $r$BC and $R_{BC}$. Campaign-average chemical composition of $r$BCc (g) and NR-PM$_1$ (h). (i) Mass ratios of species in $r$BCc to those in NR-PM$_1$ (the whiskers above and below the boxes mark the 90% and 10% percentiles, respectively; the upper and lower edge of the boxes represent the 75% and 25% percentiles, respectively; and the lines and triangles inside the boxes denote the median and mean values, respectively; SOA represents ([LO-OOA$_{r\text{BC}}$] + [MO-OOA$_{r\text{BC}}$])/([LO-OOA$_{\text{NR-PM1}}$] + [LV-OOA$_{\text{NR-PM1}}$]), and POA represents ([$r$BC-rich + HOA-rich + BBOA + WS-HOA])/HOA$_{\text{NR-PM1}}$).



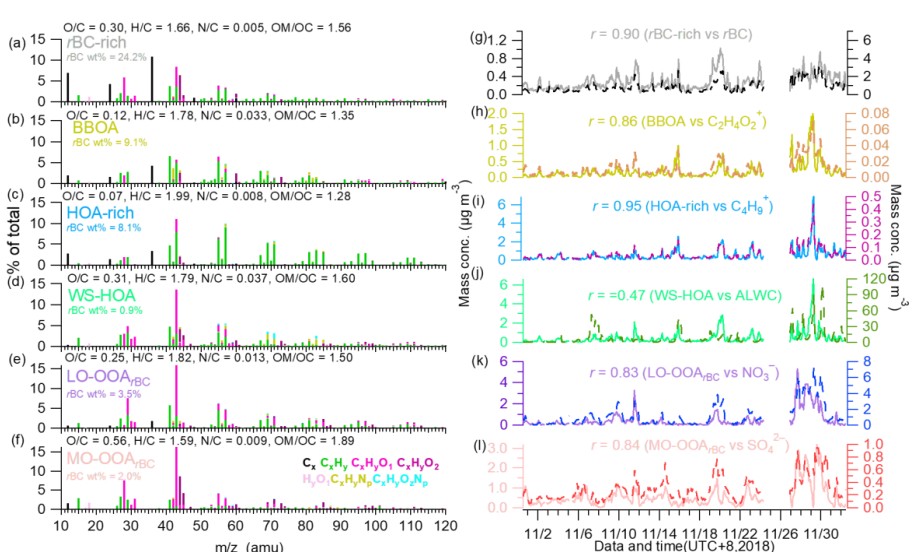

Figure 3. High resolution mass spectra of (a) $r$BC-rich, (b) BBOA, (c) HOA-rich, (d) WS-HOA, (e) LO-OOA$_{r\mathrm{BC}}$, and (f) MO-OOA$_{r\mathrm{BC}}$. (g-l) Time series of corresponding factors, their tracers ($r$BC, $C_2H_4O_2^+$, $C_4H_9^+$, ALWC, nitrate and sulfate) as well as the correlation coefficients (ALWC refers to aerosol liquid water content, which was estimated by using the extended aerosol inorganic model (Clegg et al., 1998). Calculated ALWC at different RH values is shown in Fig. S4)


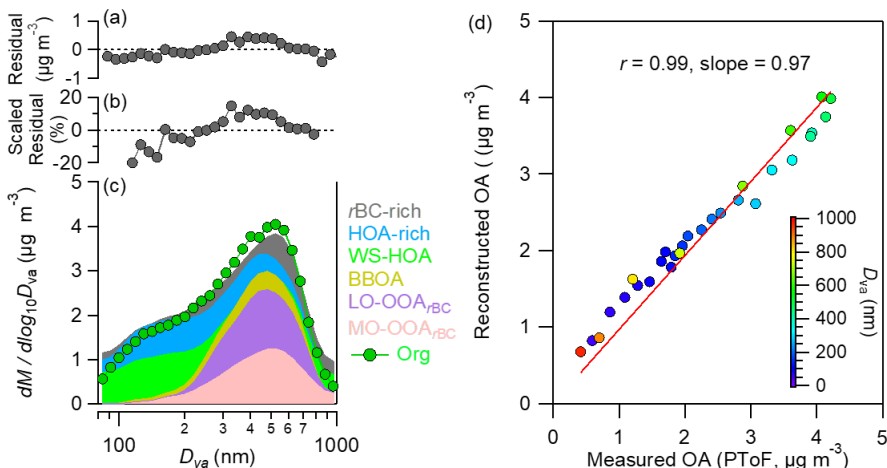


Figure 4. Summary of key diagnostic plots of derivation of size distributions of
individual $r$BCc OA factors. (a) Absolute and (b) relative residuals between the
reconstructed and measured OA mass concentrations in different size bins. (c) Stacked
size distributions of the six OA factors compared to the size distributions of total OA.
(d) Reconstructed OA mass concentrations compared to the measured values for
different size bins (80-1000 nm).



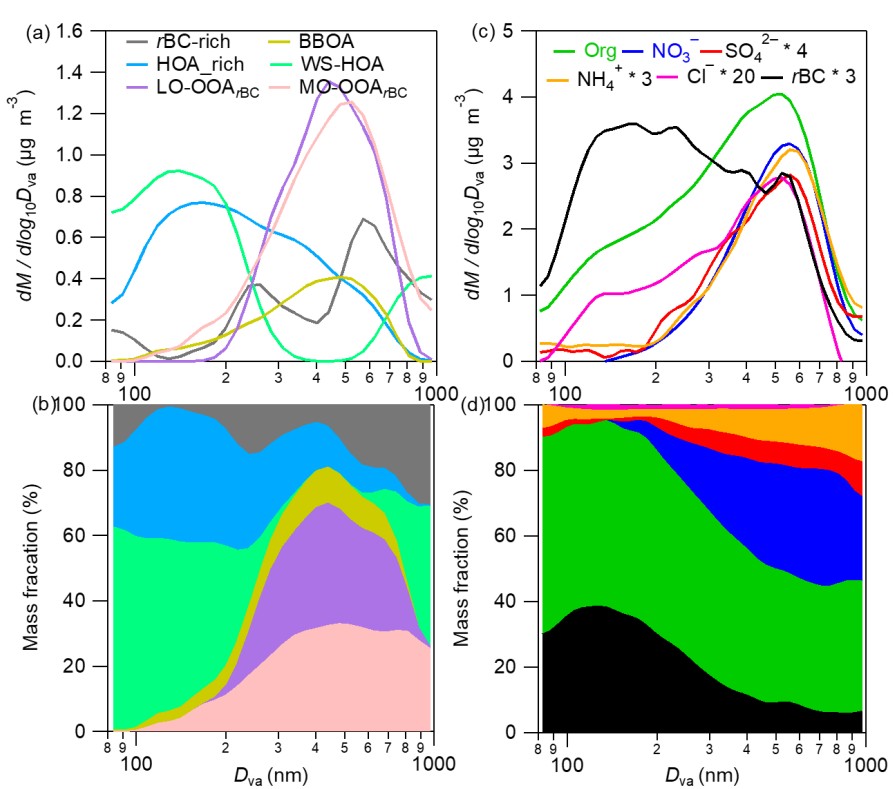

Figure 5. Campaign-average size distributions of six $r$BCc OA factors (a) and individual $r$BCc components (b), and corresponding mass contributions of the six factors to the total $r$BCc OA (c), and the major components to the total $r$BCc (d) at different sizes (80-1000 nm).




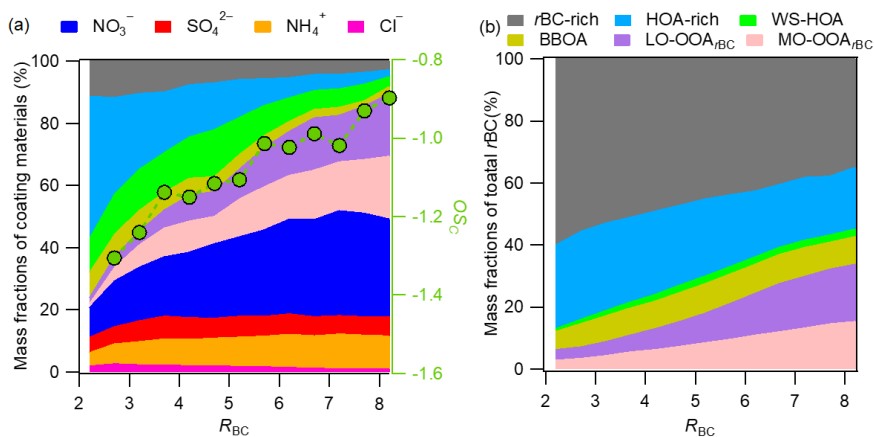


Figure 6. (a) Variations of mass fractions of the major *r*BCc components against $R_{BC}$.
(b) Variations of mass contributions of individual *r*BCc OA factors to *r*BC against $R_{BC}$.

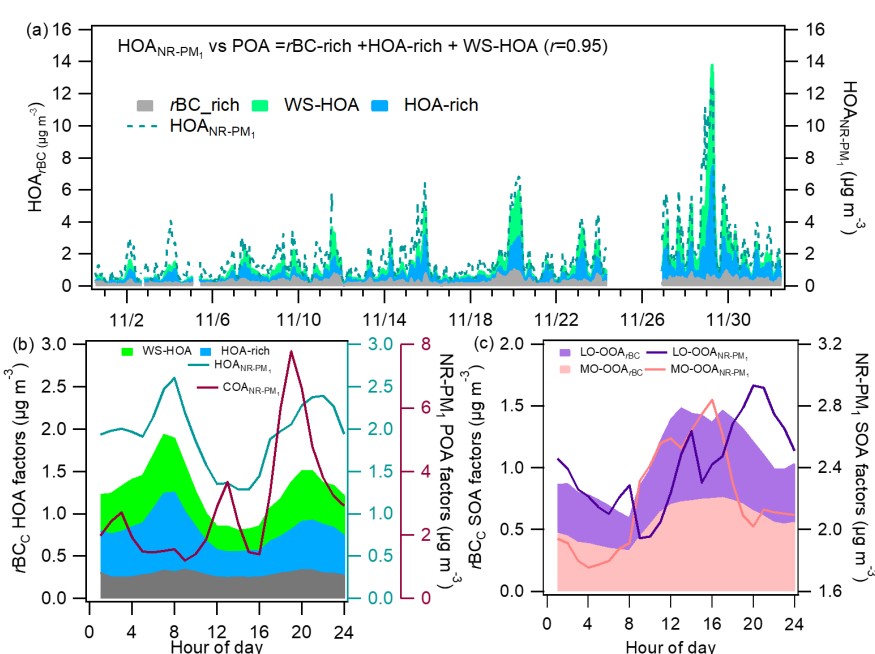

Figure 7. (a) Time series of stacked three rBCc POA factors (i.e., $r$BC-rich, HOA-rich, and WS-HOA) and $HOA_{NR-PM1}$. Comparisons of the diurnal patterns of different POA factors (b) and SOA factors (c) of $r$BCc and $NR-PM_1$.






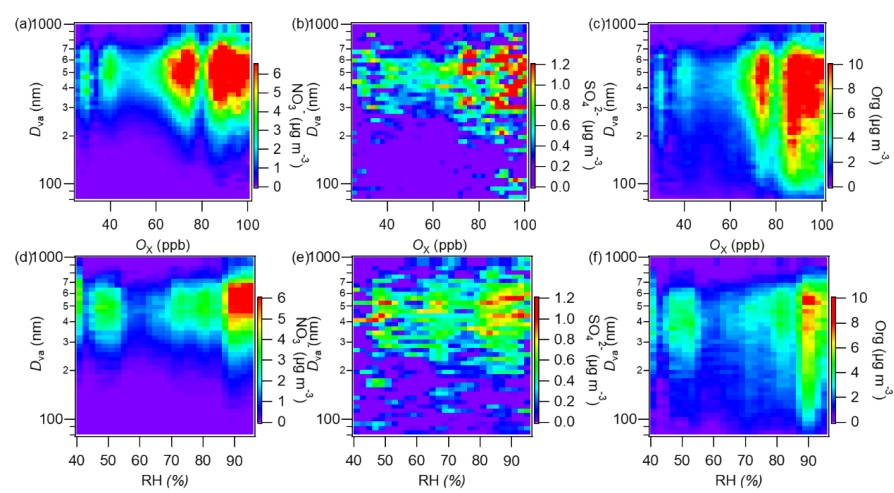


Figure 8. Image plots of size distributions of *r*BCc nitrate, sulfate, organics as a function
of (a-c) $O_X$ and (d-f) RH, respectively (color represents its concentration).





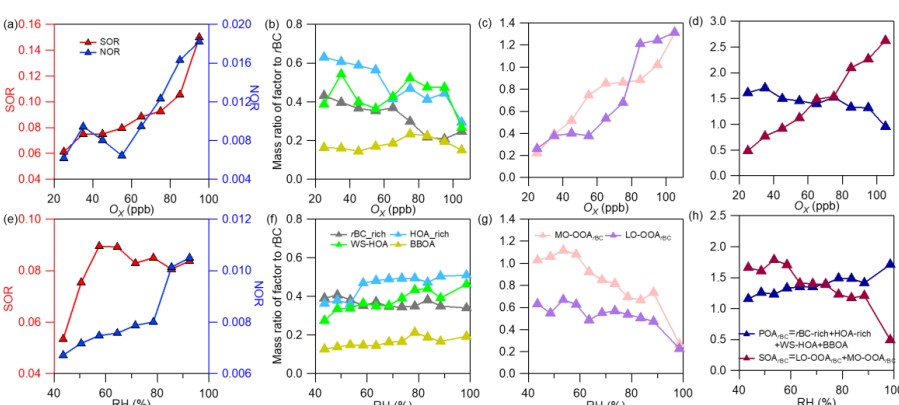

Figure 9. Variations of nitrogen oxidation ratio (NOR) and sulfur oxidation ratio, mass ratios of different POA factors, SOA factors and total POA and SOA to $r$BC against O$_X$ (a-d) and RH (e-h) (NOR= $n$NO$_3^-$/($n$NO$_3^-$+$n$NO$_2$+$n$NO) and SOR= $n$SO$_4^{2-}$/($n$SO$_4^{2-}$+$n$SO$_2$), where $n$NO$_3^-$, $n$SO$_4^{2-}$, $n$NO$_2$, $n$NO and $n$SO$_2$ are the molar concentrations of particle-phase sulfate, nitrate, gaseous NO$_2$, NO and SO$_2$, respectively).

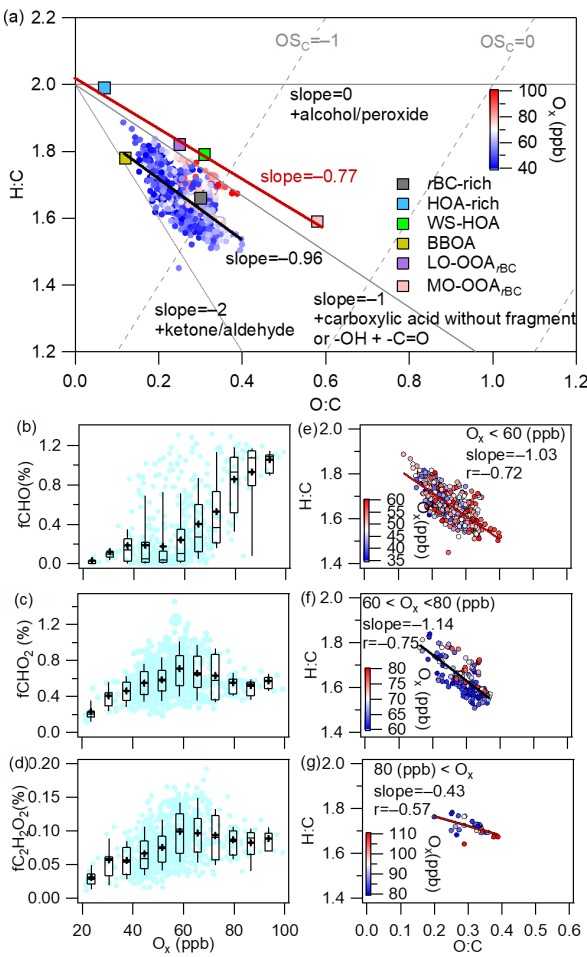

1320

Figure 10. (a) Van Krevelen diagram of H/C versus O/C ratios for all *r*BCc OA and the

six factors colored by $O_X$ concentrations (the black line represents the linearly fitted

line of all OA data, and the red line is the fitted line of the four OA factors). (b-d) Mass

fractions of selected oxygenated ion fragments as a function of $O_X$ (meanings of the

boxes are the same as those described in Fig. 2). (e-g) Scatter plots of H/C versus O/C

ratios under different $O_X$ levels (data are colored by $O_X$ concentrations).

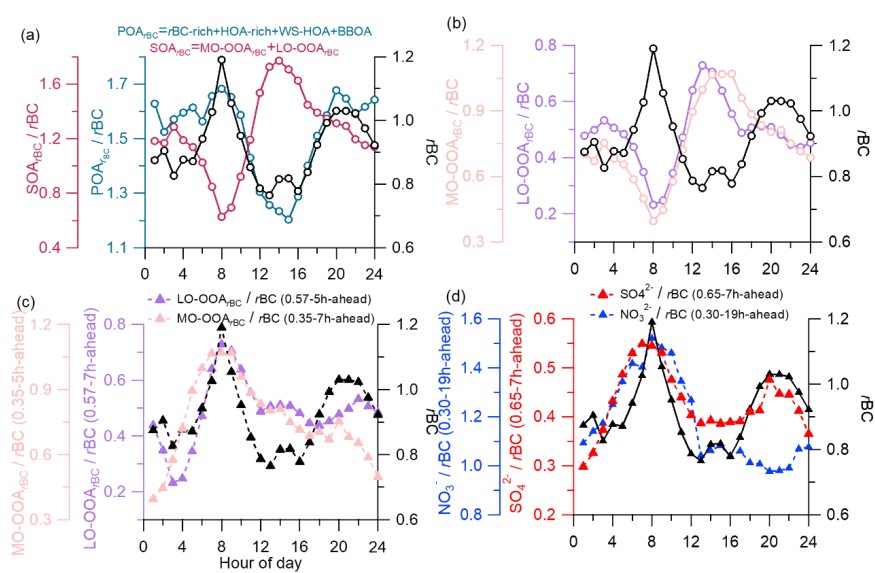

1327

Figure 11. Campaign-average diurnal patterns of (a) $r$BC, $POA_{rBC}/r$BC and

$SOA_{rBC}/r$BC, and (b) $r$BC, $MO\text{-}OOA_{rBC}/r$BC and $LO\text{-}OOA_{rBC}/r$BC. Adjusted diurnal

patterns by the average coating time (ACT) for (c) $LO\text{-}OOA_{rBC}/r$BC, $MO\text{-}OOA_{rBC}/r$BC,

and (d) $SO_4^{2-}/r$BC, $NO_3^-/r$BC.


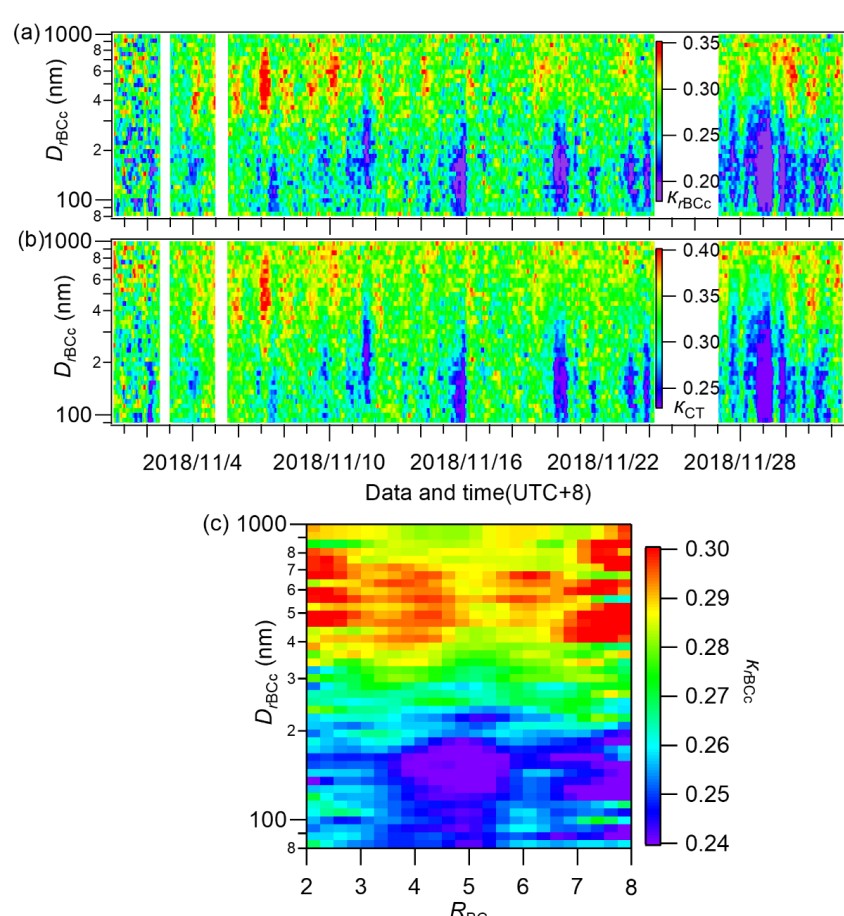


Figure 12. Image plots of size-resolved hygroscopicity parameters of (a) $r$BCc ($\kappa_{r\text{BCc}}$),

(b) its coating materials ($\kappa_{\text{CT}}$) during the whole campaign, and (c) the campaign-average

size-resolved $\kappa_{r\text{BCc}}$ at different $R_{\text{BC}}$.



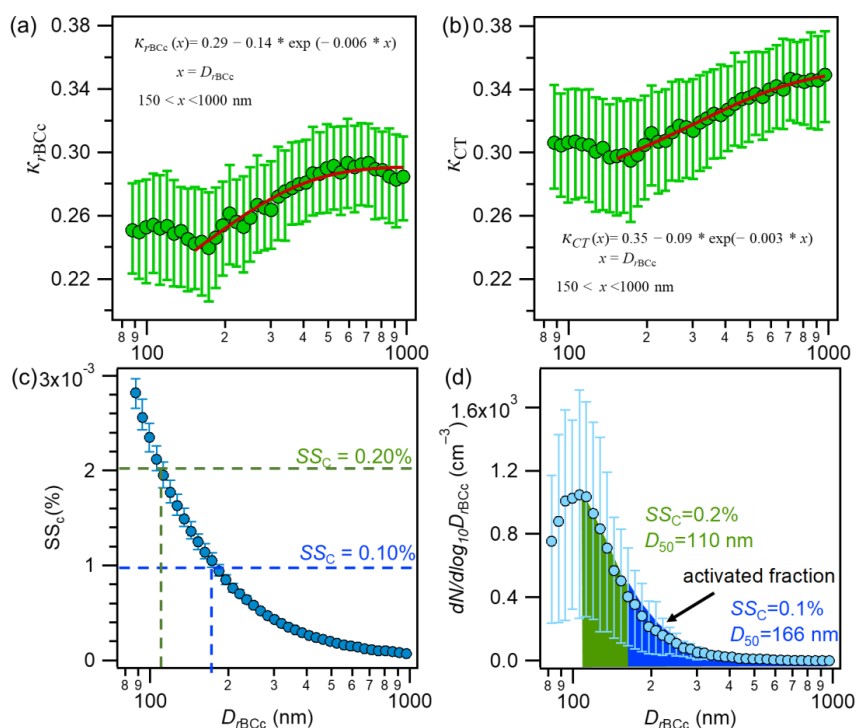


Figure 13. Campaign-average size-resolved hygroscopic parameters for $r$BCc ($\kappa_{r\mathrm{BCc}}$)
and (a) for its coatings ($\kappa_{\mathrm{CT}}$) (b) (the red lines are exponential fits of the data of 150-
1000 nm). (c) Campaign-average size-resolved critical supersaturation ($SS_C$), and (d)
the predicted activated fraction of $r$BCc number concentration based on $D_{50}$ at $SS_C$ of
0.1% (166 nm) and 0.2% (110 nm) (the solid circles are mean values, the upper and
lower lines are the 75$^{\mathrm{th}}$ and 25$^{\mathrm{th}}$ percentiles, respectively).



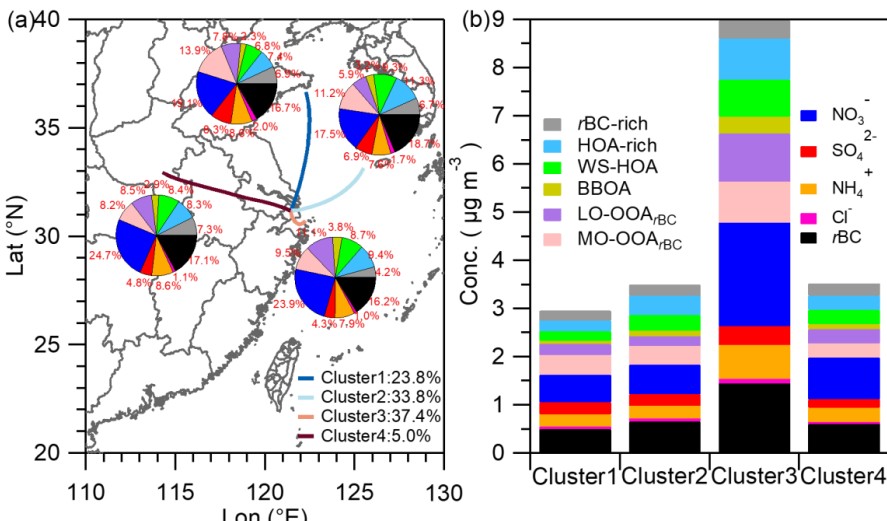

Figure 14. (a) Four clusters of 24-h backward trajectories (at altitude of 500 m) analyzed
by NOAA HYSPLIT model (http://www.arl.noaa.gov/ready/hysplit4.html) embedded
in Zefir(Petit et al., 2017), with the pie chart showing the average rBCc chemical
compositions in each cluster. (b) Stacked mass concentrations of the rBCc components
of the four clusters.

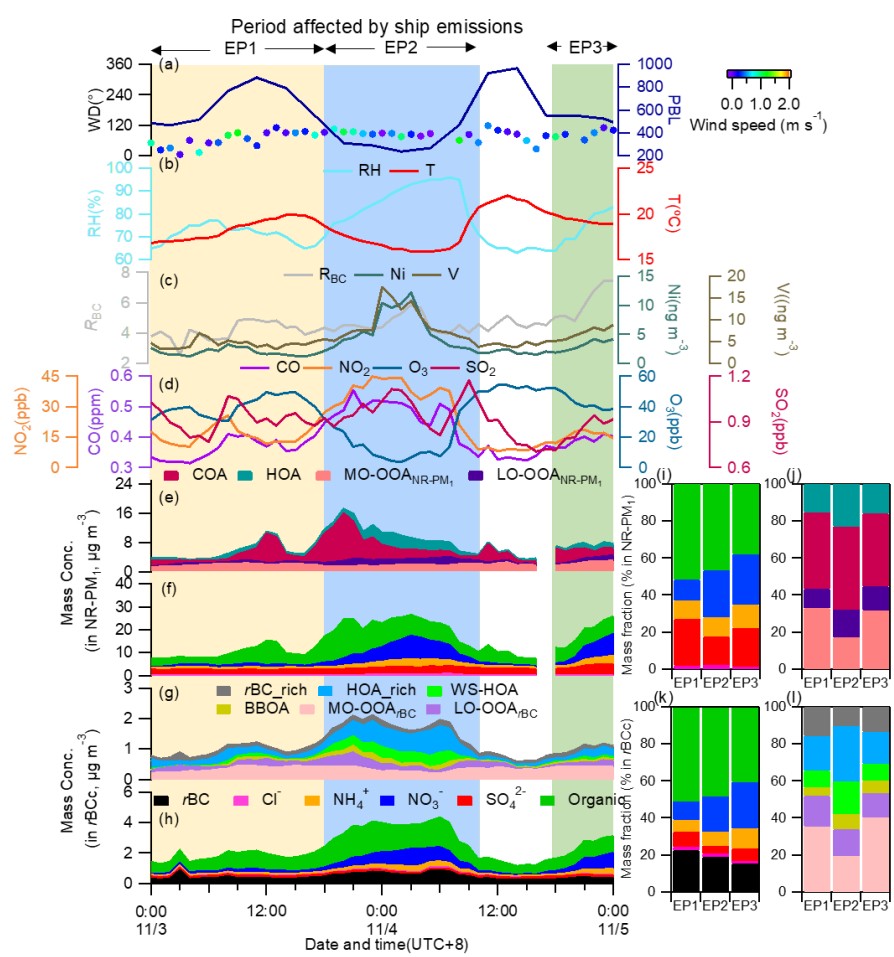

1350

Figure 15. Time series of (a) wind direction (WD) colored by wind speed (WS),
planetary boundary layer (PBL) height, (b) relative humidity (RH) and temperature (T),
(c) mass concentrations of particle-phase Ni and V, and $R_{BC}$, (d) mass concentrations of
CO, NO2, O3, SO2, stacked concentrations of (e) NR-PM1 OA factors, (f) NR-PM1
species, (g) $r$BCc OA factors, and (h) $r$BCc components during the ship emission period
(SEP). Mass contributions of different components to NR-PM1 (i), different OA factors
to total NR-PM1 OA (j), different components to $r$BCc (k), and different OA factors to
total $r$BCc OA (l) for the three episodes (EP1, EP2 and EP3).

1359