# Peer review of "Chemical properties, sources and size-resolved hygroscopicity of submicron black carbon-containing aerosols in urban Shanghai"

_Atmospheric Chemistry and Physics, 2022_

## Author Comment (AC1)

Response to comments

Dear Editor and referees:
Thanks very much for the comments regarding our manuscript. We have now addresses these comments point-by-point below with original comments in Italics, a thorough language check and modification of writing were also performed to improve the manuscript, please check the manuscript with tracked changes.

To referee #1

*This manuscript by Cui et al. investigate the submicron refractory black-carbon containing (rBCc) particle sin Shanghai, via a combination of laser-only Aerodyne SP-AMS and a HR-AMS. The former allowed the determination of rBC-containing particles exclusively and the latter was for total submicron particles. Chemical properties, sources as well as derived size-resolved hygroscopicity of the rBCc were characterized in great details. The paper is overall very comprehensive, with in-depth analyses and discussions of the rBCc properties. This reviewer finds the major findings of this work are also of importance to advance our current understanding on rBCc properties as well as its environmental impacts, in particular, the primary sources and secondary ageing processes, hygroscopicity of the coating materials. I recommend a minor revision before its acceptance, my specific comments are listed below:*

**Reply:** We thanks a lot for the referee's positive comments on our work. The specific comments were addressed below.

*PMF analysis resolved 6 factors that is well supported by relevant interpretation. I suggest to provide the mass spectra of 5 factors and 7 factors for reference (probably in the supplement) and support the best choice of 6 factors.*
**Reply:** We have added the mass spectra of 5-factor and 7-factor solutions in the supplement (new Figure S3), and added relevant description in the main text.

*ALWC calculation in Figure 3 and Figure S4 is less clear. I suggest briefly describe the E-AIM model and its procedures to obtain ALWC.*
**Reply:**More technical details regarding the calculation of ALWC was added in the supplement, and a brief description in the caption of Figure 3 was added. "The ALWC was calculated by using model II of extended aerosol inorganic model (E-AIM II), and based only on the inorganic components measured by SP-AMS. The main calculation steps are summarized below: (1) the model II determines the state of a system containing water and the inorganic salts in equilibrium at the corresponding temperature and relative humidity (converted to a value of 0-1); (2) the molar concentration of $H^+$ is obtained according to the ion balance based on the inorganics ($SO_4^{2-}$, $NO_3^-$ and $NH_4^+$) measured by SP-AMS ($Cl^-$ is not considered here as it is a very minor component), and the quantities of these four ions ($SO_4^{2-}$, $NO_3^-$, $NH_4^+$ and $H^+$) are converted to the molar concentrations in particles in per $m^3$ of air; (3) select the solids allowed to be formed according to the actual condition; (4) use these parameters (temperature, relative humidity, four ions, and allowed solids) to perform the LWC calculations using E-AIM II (http://www.aim.env.uea.ac.uk/aim/model2/model2a.php) online."

*The fact that the BBOA was resolved for rBCc but not in NR-PM1 OA, requires a bit more*

*explanation in 346-358.*

**Reply**:Thanks for the comments. We in fact explained this in Lines 449-453, and now we modified the description." "One plausible reason was that the BBOA mass contribution was minor (equivalent to <5% of NR-PM$_1$ OA mass) therefore was not able to be separated from other OA factors; another speculation is that laser only SP-AMS can detect refractory species that HR-AMS cannot, and some of these refractory species might be enriched in biomass burning OA. Identification of BBOA in $r$BCc rather than in simultaneously measured total NR-PM$_1$ was also found in Tibet and Beijing, role of such BBOA and its interplay with $r$BC core remain a subject of future work. "

*It is interesting to find that only a small portion of sulfate was coated on rBC cores while the fraction of coated nitrate on rBC was relatively large. Any educated explanation? Since sulfate had lower volatility than nitrate, this is a bit surprising.*

**Reply**:This is indeed an interesting finding. These explanation is now added in the text, "The lower fraction of sulfate in $r$BCc than nitrate was likely due to a few reasons. One probable reason is traffic was a dominant source of $r$BC (see Section 3.2.1 for details) and NO$_2$ is known to be mainly from traffic as well, therefore secondarily formed nitrate was easy to condense on co-emitted $r$BC, however SO$_2$ is mainly from other sources rather than traffic. Another possible cause is that $r$BC concentration was relatively high during nighttime, and nighttime formation of nitrate was significant; Sulfate, on the other hand, was mostly formed in the afternoon due to photochemical oxidation in this study (see Section 3.3.2 for details), whereas afternoon $r$BC concentration was low."

*Is it possible to provide the uncertainties of the regressed parameters for the expression of krBCc and kCT?*

**Reply:** We have now added the uncertainties of the obtained parameters in Figs. 13a and b.

*To referee #2*

*General comments*

*Could the authors explain why the measurement was only conducted in winter season. The higher ozone concentrations and stronger light intensity could alternate the formation of particles. In addition, the wind direction and seasonal sources could be also contributed to the formation of the BC fractions. Even the manuscript is satisfactory most presentation of the points, the language should be further polished by professional. Few non-scientific terms are being used.*

**Reply**:Thanks for the comments. The campaign was in fact conducted in late autumn and early winter. We agree that due to differences in meteorological conditions as well as other parameters (ozone concentrations, and emissions of other precursors), the secondary formation processes have seasonal differences. We in fact chose Shanghai as a model of densely populated megacity to investigate the rBCc properties, not specifically chose the season. The campaign period belonged to the cold season which often has more diverse sources and unfavorable meteorological conditions that complicate the pollution, it therefore can offer more detailed information than in other seasons, but of course measurements in other seasons are still essential for a complete understanding of rBCc behaviors. This is made clear in Section 2.1. Regarding the writing, we have conducted a careful proofread and improved the writing in particular, to avoid the use of non-scientific terms.

*More specific comments:*

*Line 243: Suggest change the presentation in format speech.*

**Reply:** Changed to "The winds with speeds <0.5 m s$^{-1}$"

*Line 267: Specify the timeline for rush hours*
**Reply:** It refers to 6-9am for the morning rush hours.

*Line 275-284: The numbers and percentages should be given with the SD. Also check them out in the entire manuscript.*
**Reply:** The SD of the values are added, and for many other values in the manuscript, the SD values were added.

*Line 428: It is not appropriate to us a non-scientific term. Please specify how is "big" defined.*
**Reply:** Agree. We deleted "but not big".

*Line 441: Do you mean the food ingredients? Replace "itself"*
Reply: Done

*Line 447: Please specify "some" here. If that is the case, how could this inference the accuracy of the results of this study.*
**Reply:** We changed to "a portion of them likely". This a possible reason that we propose, it is indeed difficult to quantify and needs further investigation.

*Line 488: Would SOR be changed while the RH is further lower? There is no comparison among a wider range of RHs. Why was no OH radical correlation determined when the secondary formation is discussed?*
**Reply:** SOR might changes when RH is further lower, but no lower RH is observed during the measurement period. We agree that if OH data is available along with the AMS measurement, it will certainly strengthen the discussion of secondary formation processes. Unfortunately, we did not have the instrument for OH measurement. This point is added in the discussion.

*Line 619: Pleas use 24-h time scale.*
*Line 703: November 3rd to 5$^{th}$*
**Reply:** Done

*Line 718: SD?*
**Reply:** SD value is added, including the concentrations in next sentence.

*Line 764: Replace "of course"*
**Reply:** Done

*Line 766: how do you define the contributions were low? Please specify.*
**Reply:** The values are specified, "(as shown in Figs. 15i and 15l, sulfate contribution was only 15.1%, and SOA contribution was only 33.7%) "

*Line 767: the lowest*
**Reply:** Done

---

## Author Response (AR2)

**Response to comments**

Dear Editor

Thanks very much for the comments regarding our manuscript. We have now addressed these comments point-by-point below with original comments in *Italics*. Please check the manuscript with tracked changes.

*Line 40, average times to "average time".*
**Reply:** Done

*It should be more clearly shown that how the calculation of "coating time" may have been affected by the boundary layer development, as it will affect the concentration of rBC, while the rBC concentration has been used in the correlation analysis.*
**Reply:** Yes, we agree that the rBC diurnal pattern could be affected by the changes of boundary layer (PBL) height. In particular, the nighttime concentrations might be elevated due to lowered PBL height. Therefore, we used the mass ratios of the species to rBC rather than mass concentration of the species to perform the calculation. The relative variations of secondary species to rBC might be a better way to reduce the impacts of PBL, and better reflect the "coating process" of the species. Of course, this is still a rough estimate of the coating process. This point is now made clear in the main text.

*I would suggest to tidy up the figures. 15 figures may be too many for this article. For example, Fig.14 could be the first figure and merged with Fig. 1. Fig. 4 - Fig. 6 could be merged as one figure but the less important information could go to supplement (also remove the duplication). Does Fig. 7 have some duplicated information with Fig. 2 and Fig. 3? Fig. 8 and Fig. 9 both conveyed RH information which may be merged. Fig. 15 contained meteorological information again. These may help the whole article look tight.*
**Reply:** Thanks for the suggestion, we agree that some figures might be redundant and should be in the supplement. Per your suggestion, Figure 14 is now directly moved into the supplement (it will be too busy to merge with Fig. 1). FIgure 4 is put into the supplement. FIgure 7a has some duplicated information (time series of POA factors) with Figure 3, and is now moved into the supplement too. As Figures 8 and 9 both contains multiple panels and it seems to be too busy to merge them together without reducing the space therefore we keep them. At last, FIgure 15 contains repeated meterological data and now the panels a and b are removed (as well as those of Figure S13). We have updated those figures and carefully checked and changed the corresponding texts. The supplement figures are renewed as well and their relevant reminders in the main text.